# NMPC-Based Workflow for Simultaneous Process and Model Development Applied to a Fed-Batch Process for Recombinant *C. glutamicum*

**Philipp Levermann** [1,†], **Fabian Freiberger** [1,†], **Uma Katha** [2], **Henning Zaun** [2], **Johannes Möller** [1], **Volker C. Hass** [3], **Karl Michael Schoop** [4], **Jürgen Kuballa** [2] **and Ralf Pörtner** [1,*]

[1] Institute of Bioprocess and Biosystems Engineering, Hamburg University of Technology, 21073 Hamburg, Germany; philipp.levermann@tuhh.de (P.L.); fabian.freiberger@tuhh.de (F.F.); johannes.möller@tuhh.de (J.M.)

[2] GALAB Laboratories GmbH, 21029 Hamburg, Germany; uma.reddy@galab.de (U.K.); henning.zaun@galab.de (H.Z.); juergen.kuballa@galab.de (J.K.)

[3] Faculty Medical and Life Sciences, Hochschule Furtwangen University, 78120 Furtwangen, Germany; hass@hs-furtwangen.de

[4] Ingenieurbüro Dr.-Ing. Schoop GmbH, 21109 Hamburg, Germany; info@schoop.de

[*] Correspondence: poertner@tuhh.de; Tel.: +49-40-42878-2886

[†] Shared first authorship.

**Abstract:** For the fast and improved development of bioprocesses, new strategies are required where both strain and process development are performed in parallel. Here, a workflow based on a Nonlinear Model Predictive Control (NMPC) algorithm is described for the model-assisted development of biotechnological processes. By using the NMPC algorithm, the process is designed with respect to a target function (product yield, biomass concentration) with a drastically decreased number of experiments. A workflow for the usage of the NMPC algorithm as a process development tool is outlined. The NMPC algorithm is capable of improving various process states, such as product yield and biomass concentration. It uses on-line and at-line data and controls and optimizes the process by model-based process extrapolation. In this study, the algorithm is applied to a *Corynebacterium glutamicum* process. In conclusion, the potency of the NMPC algorithm as a powerful tool for process development is demonstrated. In particular, the benefits of the system regarding the characterization and optimization of a fed-batch process are outlined. With the NMPC algorithm, process development can be run simultaneously to strain development, resulting in a shortened time to market for novel products.

**Keywords:** NMPC algorithm; *C. glutamicum*; model-based process development; digitalization; process optimization; process modeling

## 1. Introduction

The development of bioprocesses can be time-consuming and cost-intensive. Usually, first the production strain for the expression of the target molecule is created using molecular biology and recombinant DNA technology. After the selection of a promising production strain, the actual process development for the production of large quantities of the target molecule starts. By this step-by-step approach, usually a lot of time is wasted.

To reduce this effort, different approaches such as, for example, Design of Experiment (DoE) are used which offer a systematic method for the evaluation of multiple process variables, but still require a large number of experiments [1–3]. Furthermore, mathematical process modeling provides tremendous

potential for the characterization, design, and optimization of bioprocesses and thereby can help to reduce the amount of necessary experiments for process development [4–19]. The usage of mathematical process models in the framework of DoE approaches (model-assisted Design of Experiment, mDoE) have been described and investigated in the past. These approaches use the available process knowledge from the literature or preliminary data sets on process variables in order to simulate experiments before performing a DoE and to reduce the effort needed to define the appropriate design space. However, even if it is possible to reduce the experimental effort by this technique considerably, the required process knowledge for setting up the mathematical/kinetic model is often not available at an early stage of process development, especially if no platform technology exists [3,20–24].

In this study, a novel model-based approach is suggested which applies a Nonlinear Model Predictive Control (NMPC) algorithm as a fundamental part of the workflow. The basic idea is to design the actual bioprocess (in this study, a fed-batch) in parallel to the development of the bacterial production strain rather than consecutively. Furthermore, the number of actually performed experiments shall be kept as low as possible, and finally an acceptable process model should be at hand.

## 1.1. Setup of the NMPC

The NMPC algorithm is an adaptive model-based controller [25,26] which has been widely used for control and optimization [27–40], but hardly for process development. The usage of NMPC algorithms in this study is motivated by the success of Open Loop Feedback Optimal (OLFO) strategies [9,41,42].

The setup of the NMPC algorithm used here is based on prior studies of [6,9,43] and consists of an identification part, which estimates the states and parameters of the process model using the available experimental data, and an optimization part, which calculates, e.g., an optimal control feed trajectory for a microbial fed-batch culture based on the identified model status and a suitable optimization criterion (see Figure 1). After a certain time, when new process data become available, the whole procedure is repeated. The combined identification and optimization process is re-iterated in order to obtain an updated feed trajectory, which is subsequently passed to the process control system.

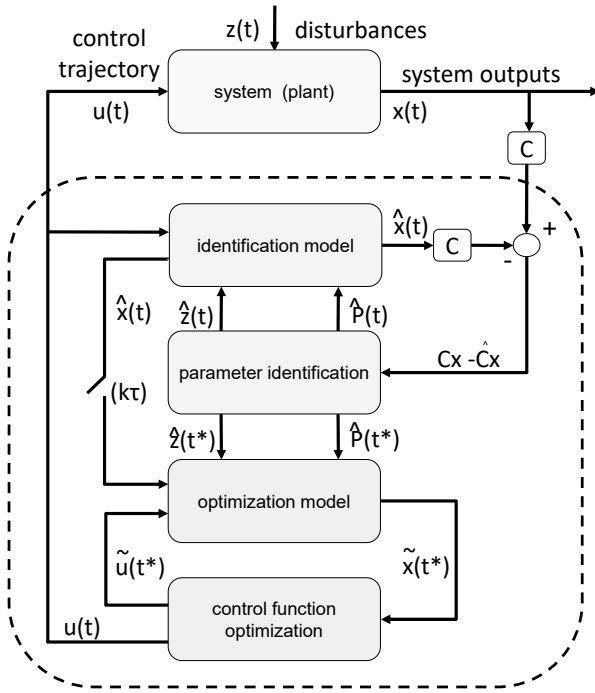

**Figure 1.** Workflow for the Nonlinear Model Predictive Control (NMPC) cycle. Followed by a model parameter estimation, the process input (here, the feed profile) is optimized (adapted from [6,9,43]).

Features of the applied NMPC process control:

- Identification of model parameters based on the current process data to get the most accurate adaption of the model to the system;
- Calculation of an optimal control function based on the actual process status;
- Controller adapts to the current process performance;
- Procedure is repeated as soon as new relevant process data are available.

The performance of such a controller depends on the model applied in the identification and optimization part [7,43]. There are requirements for the application of a model to ensure the operation of the controller in practice. On the one hand, the model has to be sufficiently complex to adequately describe the cultivation, while, on the other hand, it cannot be too complex, such that a fast and safe identification of its parameters is ensured. It has to have the ability to fit varying cultivation courses by the adaptation of its parameters from only a few measured data sets.

*1.2. Workflow of the NMPC-Assisted Process Development Strategy*

In order to use the NMPC algorithm for fast process characterization, the following workflow was formulated:

1. Existing knowledge of the process or similar processes from the literature is used to build a first process model. Additionally, initial small-scale experiments—e.g., in shaking flasks—can be performed in order to get a first process model. In general, it is recommended to start with an unstructured, unsegregated model for its robustness and to keep the computing times low. In this context, unstructured means that intracellular compounds are not considered in the model equations, so that only one state is used in the equations for the description of the biomass [44]. The term unsegregated means that the model relies on an average cell and does not consider the unique biochemical or morphological statuses of single individual cells [44]. Thus, the biomass is lumped into one homogeneous population consisting of many identical average cells. Generally, the model should be as simple as possible and as complex as needed, and it should only contain terms relevant for the process development and optimization. In that regard, it is crucial to understand possible substrate limiting and inhibiting effects by substrate(s) and metabolite(s) and to map these effects in the model.
2. When a promising model is found, the next step is to run an NMPC-controlled fed-batch in a bioreactor. This is either done successfully or until failure. Failure in this context means that the simulated cultivation courses do not reflect the measured data sufficiently, which indicates an insufficient model. Success is reached, if the outcome of the fed-batch is actually in coincidence with the beforehand-defined target—e.g., maximized biomass concentration or product yield—indicating that the process is already close to the optimized process.
3. If the NMPC-controlled fed-batch fails at some point, the process is not understood well enough. In this case, additional experiments and analytics are needed in order to improve the process understanding and to extend the model accordingly. Therefore, different tools can be used, varying from laboratory analytics of different byproducts and metabolites or even a mDoE approach. Byproducts or metabolites can lead to inhibiting effects which occur at some time point of the process.
4. This procedure is repeated until the process is optimized sufficiently and a standardized optimized procedure can be derived. At this point, the NMPC can still be used passively as a monitoring tool for the process.

In Figure 2, a schematic overview of this approach is given. Please keep in mind that the main purpose of this approach is not to generate a deeper mechanistic understanding of the investigated process, but rather to design the process in an efficient manner.

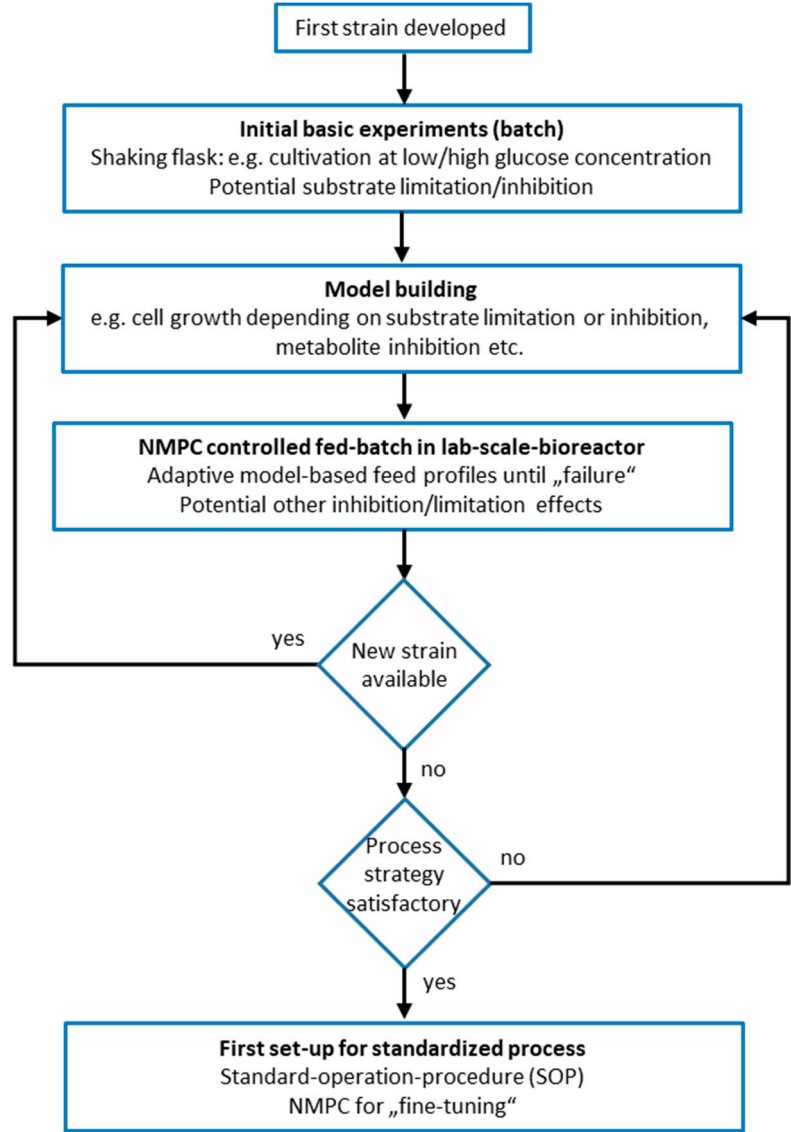

**Figure 2.** Schematic overview of the NMPC-assisted workflow for process development.

Biological processes tend to have, to some degree, an unpredictable nature in general. A model predictive control approach which uses the current process data for process optimization as described above can overcome occurring peculiarities and disturbances. In detail, on-line and at-line process data allow a model parameter identification to get the most accurate adaptation of the model for the running process. Based on this model, it is possible to predict the future process. For the presented case of a fed-batch process, the feed trajectory was chosen as a process input for optimization. Process extrapolations allow the NMPC algorithm to calculate the optimized feed trajectory. As stated above, the whole procedure is repeated in a fixed circle time (moving prediction horizon). Thus, the model and the feed profile are readapted constantly. Therefore, it is possible to optimize the process while it is running and to react to uncertainties accordingly.

As for a fed-batch process, the NMPC algorithm is used to control the feed profiles of the process; it is crucial to understand the dependency of the reaction rates on the substrates. This dependency, in the form of reaction rate limiting and inhibiting terms, has to be implemented in the model in order to control the fed-batch process.

Using the NMPC algorithm as a control for bioprocesses allows us not only to control the process, but also its characterization while it is running. This leads to the possibility of using the

NMPC control as a process development tool. This strategy aims to develop the process as fast as possible with respect to the pre-defined evaluation criteria—e.g., maximized biomass concentration or product yield. Following this strategic approach, it is possible to develop the strain and the process itself simultaneously. This already shortens the required process development time tremendously. Furthermore, as the mathematical model is improved during the consecutive rounds, a deeper understanding of the investigated process is obtained.

The NMPC algorithm was completely implemented in the control software. As a result, the whole NMPC algorithm can be automated. All in all, an NMPC-assisted strategy has been established which allows the fast development, characterization, and control of nearly any given microbial process.

### 1.3. Goal of the Chosen Corynebacterium glutamicum Process

To illustrate the potential of this approach for process development and the characterization of bioprocesses, the workflow was performed for a *C. glutamicum* process. The aim of this process is to produce the milk oligo saccharide (MOS) 2'-Fucosyllactose (2'-FL), which acts as a prebiotic and probiotic [45–48]. During strain development, four 2'-FL producing strains were generated which were used in the NMPB-based process development strategy. Due to patent issues, further details of the engineered strains cannot be disclosed at this time point.

## 2. Materials and Methods

### 2.1. Implementation of the NMPC within the Control System

A bioreactor setup entirely controlled by the control software WinErs (IB Schoop GmbH, Hamburg, Germany) with the implemented NMPC algorithm was established. This way, in addition to the commonly used control system for the bioreactor (controls for stirrer speed, pH-value, temperature, etc.), the model system for the process extrapolation and optimization is also realized in the control software. In Figure 3, a schematic overview of the setup is given. The right side shows the bioreactor control system which controls and measures all the relevant process data. The left side depicts the model system, which is connected to the control system but realized on another computer such that the control system is independent and secured from possible interruptions. The model system retrieves the measured data from the control system via IP connection and can transfer feed profiles to the control system.

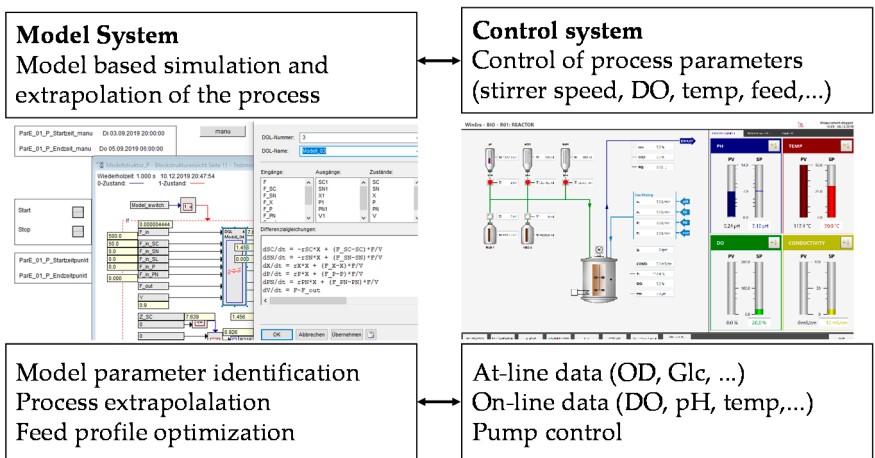

**Figure 3.** Schematic overview of the used control system setup. Model system and process control are realized together in one control system.

## 2.2. NMPC-Based Workflow for C. glutamicum Process

In the case of the investigated *C. glutamicum* process, the NMPC algorithm is used to control the feeding rate of a fed-batch process with the target of maximizing the biomass concentration within a fixed period of time. For this purpose, the following workflow was executed.

### 2.2.1. Model Adaptation—Parameter Estimation

The actual process is started as batch. After the exhaustion of the substrate (mainly glucose, indicated by a sudden increase in the dissolved oxygen concentration (DO)), feeding is started.

Samples for the determination of biomass and glucose concentration were taken every hour. After the analysis of the samples, the results were entered into the control software as sample data. A Nelder–Mead algorithm [49] integrated in the control system was used for the identification of the model parameters. The first parameter estimation was performed by the end of the batch phase in order to calculate the first feed trajectory (see Section 2.2.2). After the start of the feed phase, the model parameters were readapted every hour. All the available data of the particular fed-batch experiment available at the current time frame were used for the process model adaptation (estimation horizon).

The target function to minimize is given by the $L^2$-norm of the sample points relative to the simulated points for both biomass and glucose.

$$f_{\min} = \sum_{i=1}^{n} \left(X_{\exp,i} - X_{\sim,i}\right)^2 + \sum_{i=1}^{n} \left(S_{\exp,i} - S_{\sim,i}\right)^2. \tag{1}$$

Here, $X_{\exp}$ and $S_{\exp}$ denote the experimental sample points for biomass and glucose, and $X_{\sim}$ and $S_{\sim}$ denote the corresponding simulated data points. n indicates the number of available sample points and increases during the fed-batch in each optimization circle by one.

The objective of the control is to maximize the biomass concentration by optimizing the glucose feeding rate. Therefore, these two states are chosen to be the relevant states for the model parameter adaptation. The reasons the glucose concentration and biomass concentration were chosen to be equally weighted for model fitting were the following: First of all, the biomass concentration and glucose concentration samples were taken in constant time intervals (one hour), such that a weighting regarding differing time gaps is not needed. Consequently, the number of available data points for each are the same (denoted by n). Secondly, both are of the same order of magnitude, varying from 1 to 50 g $L^{-1}$. Therefore, it is sufficient to use absolute deviations of instead of, for example, relative deviations. For other processes, the formula can be adjusted accordingly.

In order for the Nelder–Mead algorithm to not get stuck in a local minimum, it is crucial to start with reasonable starting values for the parameter adaptation. In our case, the results of the preceding experiment were used as starting vector for the first parameter estimation. In the following NMPC cycles, the parameter estimation result of the previous cycle was used as the starting vector. A termination tolerance of $10^{-4}$ was used as a convergence criterion applied to both the change in the target function value as well as the change in the parameter vector. On average, 50–500 iteration steps were required by the solution algorithm.

Based on the adapted model it is possible to optimize the process in the following manner.

### 2.2.2. Process Optimization—Feed Trajectory Calculation

Based on the adapted model, it is possible to simulate the future course of the process with respect to a certain feeding rate. In this next step, the Nelder–Mead algorithm is used to optimize this feeding rate, such that the maximized biomass concentration at the end of an extrapolated one-hour time window is reached. In detail, the optimization algorithm varies the feeding rates and extrapolates the cultivation courses over the next hour. The target function to minimize is given by the inverse of the

simulated state for the biomass concentration at the end of the extrapolated 1 h time window starting at a given time point $\tau$ and ending at $\tau + 1$ h (moving prediction horizon):

$$f_{min} = \frac{1}{X_{sim}(\tau + 1\,h)}. \tag{2}$$

These two steps (Sections 2.2.1 and 2.2.2) are repeated every hour. This NMPC procedure of parameter estimation followed by the calculation of a new feed trajectory is schematically shown in Figure 1.

Due to the structure of the model with terms describing the substrate limiting and inhibiting effects on the growth, it can be assumed that there is only one minimum for the optimal feeding rate, provided a reasonable choice of model parameters was achieved. As for the parameter estimation, a termination tolerance of $10^{-4}$ was used as the convergence criterion.

For illustration, the whole workflow of sample analytics, followed by the process optimization, respectively, of the parameter estimation and feed trajectory calculation is shown in Figure 4.

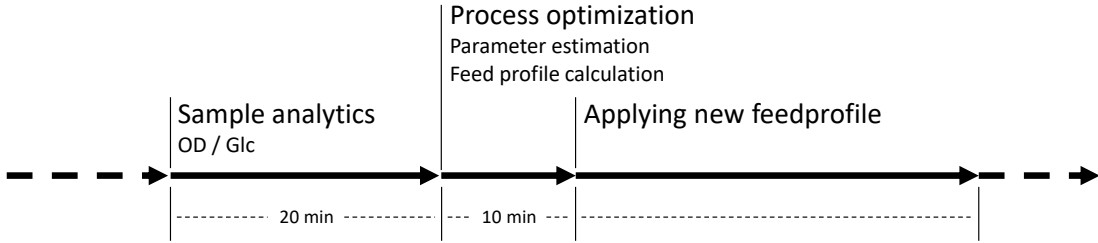

**Figure 4.** Schematic workflow to apply the NMPC control to a fed-batch process with a cycle time of one hour.

As sample analytics, the parameter estimation and feed profile calculation take approximately 30 min and the process needs to adapt to the actual feeding rate for at least 30 min; a one-hour step width (prediction horizon) for the NMPC control is the shortest step width which allows a stable automated NMPC-controlled process sequence.

### 2.3. Experimental Setup

During the project, four recombinant *C. glutamicum* strains were generated. For maintenance and cultivation, the following protocol was used. Strains were stored at −80 °C on cryo glass beads. For thawing, one cryo bead was added to a 250 mL shaking flask containing 30 mL of Lysogeny Broth (LB medium, Lennox) (Roth, Karlsruhe, Germany). The culture was then incubated at 30 °C while shaking at 220 rpm. For the acquisition of growth parameters, batch cultivations were carried out in 250 mL shaking flasks with 40 mL of CgXII medium (see Table A1), which was inoculated to an $OD_{600}$ of 1 with the LB culture. The inoculum for the fed-batch cultivations was prepared by following the batch cultivation procedure.

All the fed-batch cultivations were performed in a 2 L stirred tank bioreactor (MDX Biotech, Nörten-Hardenberg, Germany) with an initial culture volume of 1 L. MOPS-free CgXII medium was used in all bioreactor cultivations. The cultivation broth was tempered to 30 °C by a water bath. By the addition of 2.5 M of $H_2SO_4$ and 25% ammonia, the pH was controlled to 7. The DO level was kept above 20% by an automatic increase in the agitation rate or a manual increase in the aeration rate, respectively. Foam formation was inhibited by the stirrer setup, consisting of a flat blade impeller at the lower end and a pitched bladed impeller at the upper end of the agitator shaft. An additional impeller was mounted in the gas phase of the reactor for foam breaking. Then, 100 µL of Struktol J647 (Schill and Seilacher, Hamburg, Germany) per liter of culture volume was added 5 min after the inoculation, with an initial $OD_{600}$ of 0.3 to 0.5.

The optical density at 600 nm ($OD_{600}$) was measured in a conventional photospectrometer. The clucose concentrations were determined enzymatically with the YSI 2900-D (YSI Incorporated, Yellow Springs, OH, USA).

For the feed, a solution containing $500\,\mathrm{g\,L^{-1}}$ of glucose and $50\,\mathrm{g\,L^{-1}}$ of yeast extract (Ohly, Hamburg Germany) was prepared. Both components were sterilized by autoclaving separately. The feeding rates were adjusted according to the NMPC control.

Product formation was induced about one hour before the end of the batch phase. Due to legal disclosure, no further details can be presented.

## 3. Results and Discussion

In the following, first the results of the initial experiments and thereafter the NMPC-controlled fed-batches are shown to illustrate the development of the *C. glutamicum* process. The process development strategy outlined in Section 1.2 was followed. In Table 1, the performed experiments are listed with the corresponding strain generation and used model. For every strain, one experiment was planned and executed.

**Table 1.** List of the experiments with the corresponding strain generations and used models.

| Experiment Number | Strain Generation | Model | Product Formation |
|:---:|:---:|:---:|:---:|
| 1 | strain 1 | A/B | - |
| 2 | strain 2 | B(+) | - |
| 3 | strain 3 | C (with weighted sample points) | - |
| 4 | strain 4 | D (with weighted sample points) | + |

The process and the strain were developed simultaneously, such that the process had to be adapted to the new strain generations consecutively. Accordingly, the model was built up and extended consecutively over the course of the performed experiments. In the following, the model equations are shown. Thereafter, the time courses of the cultivations listed in Table 1 are discussed.

### 3.1. Model Equations

The models consist of mass balance equations and kinetic equations and were established successively in the presented workflow, which is discussed below. After investigating the appropriate models from the literature [7,50–56], as a starting point Model A was chosen.

In Model A, the cell-specific growth rate $\mu$ depending on the substrate concentration is expressed by a Monod kinetic (see Table 2, first column). For Model B, a substrate inhibition term is added (see Table 2, first and second column). In Model B+, the parameter values for the parameters $\mu$ and $Y_{XS}$ for the batch and feed phase vary. The equations for Model B and Model B+ are essentially the same (see Table A7). In Model C, the cell-specific growth rate $\mu$ has a Monod term; a substrate inhibition; and an inhibiting metabolite, which is only present in the feed phase (see Table 2, first three columns). The term for the metabolite was added in order to diminish the growth rate in the feed phase and is not verified or determined analytically. It is an artifice to take the lower growth rate at the later stages of the fed-batch into account.

In Model D, the states and balance equations for an inducer and product formation are added (see Table 2, all four columns). The product formation rate $q_P$ is proportional to the growth rate $\mu$, such that a growth-coupled product formation is simulated. The growth-coupled product formation is in coincidence with the experimental observation and is verified by the simulation.

All the used models are shown in Table 2. For a distinctive presentation of the used models, see Table A3 (Model A), Table A5 (Model B), Table A7 (Model B+), Table A15 (Model C), and Table A19 (Model D).

**Table 2.** Model equations for NMPC. The left side shows the equations for Model A. In the middle-left, the additional inhibition term for Model B is listed, and the middle-right side shows the additional equations for Model C. All equations together form Model D, which also takes the product formation into account.

| | A | B | C | D |
|---|---|---|---|---|
| | *Balance equations* | | | |
| $\frac{dX}{dt} =$ | $\mu \cdot X - \frac{F}{V} \cdot X$ | | | |
| $\frac{dS}{dt} =$ | $-\frac{1}{Y_{XS}} \cdot \mu \cdot X + \frac{F}{V} \cdot (S_0 - S)$ | | $-\frac{1}{Y_{MS}} \cdot q \cdot X$ | $-\frac{1}{Y_{PS}} \cdot q_P \cdot X$ |
| $\frac{dM}{dt} =$ | | | $q \cdot X - \frac{F}{V} \cdot M$ | |
| $\frac{dL}{dt} =$ | | | | $-\frac{1}{Y_{PL}} \cdot q_P \cdot X - \frac{F}{V} \cdot L$ |
| $\frac{dP}{dt} =$ | | | | $q_P \cdot X - \frac{F}{V} \cdot P$ |
| $\frac{dV}{dt} =$ | $F_{in} - F_{out}$ | | | |
| | *Kinetic equations* | | | |
| $\mu =$ | $\mu_{max} \cdot \frac{S}{(S+K_s)}$ | $\cdot \frac{I_S}{(S+I_S)}$ | $\cdot \frac{I_M}{(M+I_M)}$ | |
| $q =$ | | | $q_{max} \cdot \frac{\mu}{\mu_{max}}$ with $q_{max} = 0$ for $t < t_{Feed,Start}$ | |
| $q_P =$ | | | | $q_{P,max} \cdot \frac{\mu}{\mu_{max}} \cdot \frac{L}{(L+K_L)}$ |

### 3.2. Cultivation Data

In the following, the experimental data together with the simulation results are discussed in the context of the process development workflow.

#### 3.2.1. Experimental Setup 1-Strain 1–Batch and Fed-Batch Model A/B

Following the process development strategy (see Figure 2), first initial shaking flask experiments were performed. The shaking flask experiments had varying initial glucose concentrations of 5, 8, 12, and 16 g L$^{-1}$ in order to study the substrate limiting effects on growth. Based on these data, the model parameters for Model A were estimated. The results are listed in Table A4. In Figure 5, the experimental and simulated results are plotted against each other. The experiment and simulation are in good agreement with each other.

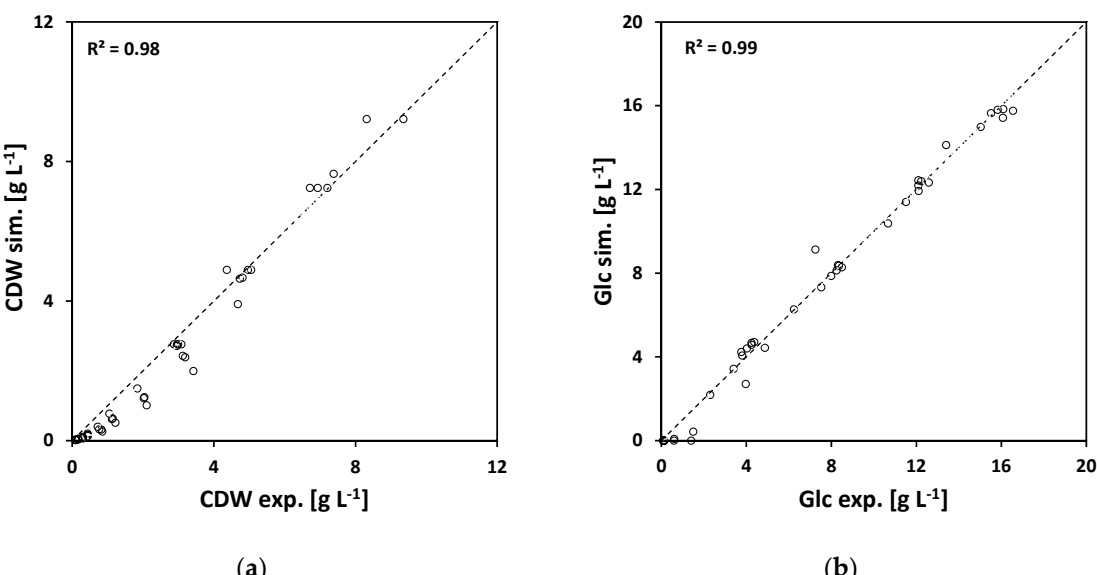

**Figure 5.** Quality of fit (Model A) for the shaking flask experiments with varying initial glucose levels of 5, 8, 12, and 16 g L$^{-1}$ for cell dry weight (**a**) and glucose (**b**) (conversion OD$_{600}$ 1 ~ 0.39 g L$^{-1}$).

The initial shaking flask experiments led to a first description of the process regarding the substrate-limiting effects. In order to control the feed trajectory with an NMPC, the underlying model also has to have some form of substrate inhibition term. Otherwise, the feed optimization would be somewhat trivial. Hence, a substrate inhibition term was added and the model was extended into Model B (Table 2, Table A7). To study the substrate-inhibiting effects, a fed-batch with a constant feeding rate was performed. The experimental and simulated cultivation courses are shown in Figure 6. The results of the parameter estimation are listed in Table A6. This experiment was also used to study process parameters such as inoculum density (OD $\geq$ 0.3), aeration rate (0.5–1.0 vvm), and stirrer speed. All the estimated parameters for these first experiments can be found in Section Appendix A.1.

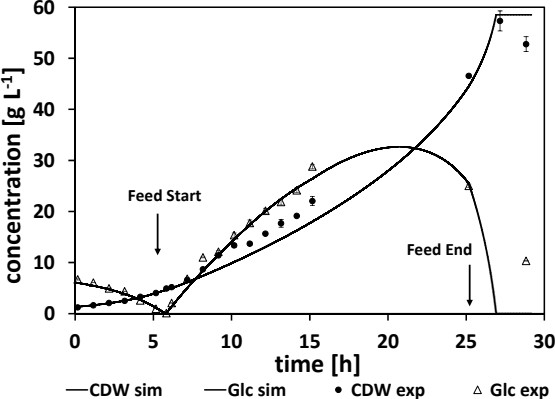

**Figure 6.** Time course of the glucose and biomass concentration for fed-batch 1; symbols mark samples; the samples were measured as three technical replicates; lines mark the adapted simulated cultivation course (Model B); constant feeding rate of 12 mL h$^{-1}$.

### 3.2.2. Experimental Setup 2-Strain 2–NMPC Fed-Batch Model B+

After the first initial experiments with strain generation 1 had been performed and the design of Model B was finished, the first two steps of the process development strategy (see Section 1.2) had been accomplished. Step 3 of the process development strategy (Figure 2) is the execution of an NMPC-controlled fed-batch. This step was performed with strain generation 2 and Model B+.

To illustrate the workflow of an NMPC-controlled fed-batch experiment, the experiment is discussed in more detail in the following. By the end of the batch phase of the experiment (when the glucose levels start to drop), the model parameters were adapted to the experimental data (see Section 2.2.1 for a detailed explanation of the parameter estimation). For this experiment, this step was performed at a process time of about t = 8 h. Based on the adapted model, the time point for the complete consumption of glucose (time point when the glucose reaches 0 g L$^{-1}$, named "drop time" in the following) is calculated by process extrapolation with the adapted model. In the second step, the optimal feeding rate beginning at the calculated drop time is determined (see Section 2.2.2 for a detailed explanation of the feed optimization) with the target to maximize cell growth. The result of this parameter estimation and the corresponding feed trajectory optimization is listed in Table A9. The calculated drop time was at about $\tau$ = 8.5 h and at the calculated initial feeding rate of 42 mL h$^{-1}$. In Figure 7a, the state of the experiment at that moment is shown. The experimental data for the cell dry weight and glucose are shown as marks and the simulated cultivation courses are shown as lines. On the left side of the vertical dashed lines is the time frame where the process data for the parameter estimation are collected from. It is denoted as the estimation horizon (EH) of the NMPC algorithm. In between the vertical dashed lines is the prediction horizon (PH) of the NMPC algorithm. Here, the calculated feed profile of $F_{in}$ = 42 mL h$^{-1}$ is shown, starting at about $\tau$ = 8.5 h, and the resulting extrapolated simulation courses the for biomass and glucose concentration are displayed. In this experiment, the feed started as the DO signal increased, indicating glucose depletion. This was at process time t = 9 h, and thereby half an hour after the predicted time point.

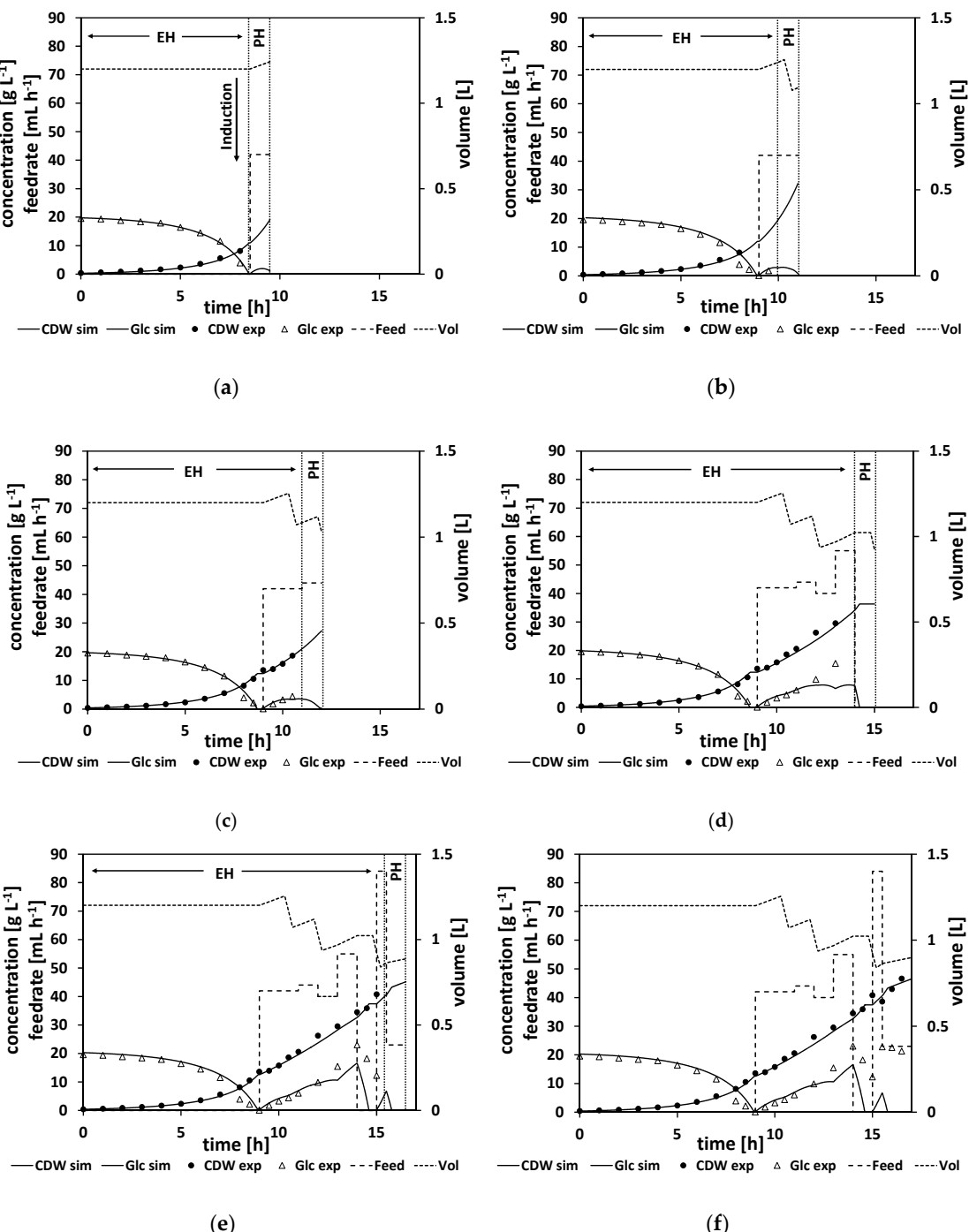

**Figure 7.** Time course of the glucose and biomass concentration for fed-batch 2. Left side of the vertical dotted lines indicate the estimation horizon (EH): symbols mark samples, the samples were measured as three technical replicates, lines mark the adapted simulated cultivation course (Model B+), the dashed line shows the calculated feed profile, the dotted line marks show the simulated reactor volume. Between the vertical dotted lines indicating the prediction horizon (PH): the dashed line shows the optimized feed profile, the lines mark the extrapolated cultivation course, the dotted line marks the simulated reactor volume. (**a**) Prediction of the starting time point for feed and the calculation of the optimized first feed rate with extrapolation starting time point $\tau = 9$ h. (**b**) Calculation of the optimized second feed rate at $\tau = 10$ h. (**c**) Calculation of the optimized third feed rate at $\tau = 11$ h. (**d**,**e**) Calculation of the optimized sixth and seventh feed rate at $\tau = 14$ h and $\tau = 15$ h. (**f**) Finished process. For the parameter estimation results and the feed optimization results, see Tables A8–A10. For model equations, see Table A7 or Table 2.

As outlined in Section 2.2 the whole NMPC procedure of parameter estimation and feed profile calculation is repeated every hour. Therefore, at a process time close to t = 10 h, the model parameters were readapted with regard to the new process data from sample analytics. Based on this newly adapted model, the next feed trajectory was optimized beginning at $\tau$ = 10 h. The results of this parameter estimation and the resulting optimized feed profile of $F_{in}$ = 42 mL h$^{-1}$ are listed in Table A10. The corresponding cultivation courses at that time point are shown in Figure 7b, with the estimation horizon on the left side of the vertical dashed lines showing the experimental data and the simulated cultivation courses fitted to this data and the prediction horizon in between the vertical dashed lines, with the optimized second feeding rate and the corresponding simulated courses for the process states, biomass concentration, glucose concentration, and volume.

This whole procedure was repeated every hour. In Figure 7c, the state of the experiment is shown when the third cycle of the NMPC algorithm was performed at a process time close to $\tau$ = 11 h. The third feeding rate was calculated based on the model with the once again, for the third time, adapted model parameter set (see Table A10 for the adapted model parameters and the calculated feeding rate values).

Figure 7d and e show the cultivation at even later stages of the experiment, and Figure 7f shows the finished process. Note that the estimation horizon grows for every subsequent NMPC cycle and more and more process data become available for parameter estimation, while the prediction horizon remains a one-hour time window and is moving further. All the estimated parameters and the calculated feed trajectory for this experiment can be found in Section Appendix A.2 in Tables A8–A10.

When studying the resulting feed trajectory and comparing the different feeding rates, keep in mind that the volume of fermentation broth is changing over the time course. In order to prevent foam building in the bioreactor, the fermentation volume was lowered from time to time in this and all the following experiments.

It is also noteworthy that Model B was adjusted while the process was running in the following way. After the calculation of the first feeding rate, the reaction rate $\mu_{max}$ and the parameter $Y_{XS}$ were separated into two parameters $\mu_{max,1}$ and $Y_{XS,1}$ for the batch phase, as well as $\mu_{max,2}$ and $Y_{XS,2}$ for the feed phase. This simple modification was made in order to take the slower growth rate in the feed phase into account. The corresponding model is Model B+ (see Table A7). The model equations are essentially the same as in Model B. While, in the batch phase, the maximum specific growth rate was about $\mu_{max,1} \approx 0.6$ h$^{-1}$, for most NMPC cycles it reduced its value by half in the fed-batch phase to about $\mu_{max,2} \approx 0.3$ h$^{-1}$ (see Table A10).

In this experiment, a biomass concentration of 42 g L$^{-1}$ was reached, which corresponds to an $OD_{600}$ of about 108, which is already a relatively high cell density. Noteworthy here are the relatively high measured glucose levels, especially at the end of the feed phase. This indicates that the model predicted too-high consumption rates, leading to too-high feeding rates. Ideally, the NMPC should keep the glucose concentration relatively low, at about $\sqrt{K_S I_S}$ (for our parameter values, at about 1.6 g L$^{-1}$), which is the glucose level that would maximize the growth rate according to the model structure. As can be seen in Figure 7d–f, the simulation cannot reproduce the high glucose levels measured at the later stages of the process, leading to too-high feeding rates to balance out the too-low simulated glucose levels. Additionally, the growth seems more linear than exponential in the feed phase. This linear growth cannot be simulated accordingly by the model. All in all, the NMPC predicted too-high consumption rates compared to reality, and therefore Model B, although leading to a high CDW concentration, can still be improved to get a more accurate controlled process.

### 3.2.3. Experimental Setup 3-Strain 3–NMPC Fed-Batch Model B+ (Weighted Sample Points)

Following the process development strategy (see Section 1.2 step 3 → step 2 and Figure 2), in the model the NMPC control has to be adjusted according to our results from the previous NMPC-controlled fed-batch experiment 2. As analyzed above, especially at the later stages of the experiment, the simulation could not fit the experimental data sufficiently to ensure the calculation of

a reasonable feed trajectory. In order to solve this problem, the model remained unchanged, but the way the parameter estimation procedure was adapted by introduction of weights.

More precisely, in order to get a more accurate adaptation of the simulation to the experiment, at the later stages of the experiment the sample points were weighted, such that the last available sample point (the newest data) weighs 10 times as much as the first available sample point within each estimation horizon. This modification was made to get a closer adaptation of the model to the process at later stages of the experiment in order to get a more realistic extrapolation and thereby a better feed profile calculation. The weighted target function for the parameter identification is given by:

$$f_{min} = \sum_{i=1}^{n} \alpha(i) \cdot \left( X_{exp,i} - X_{sim,i} \right)^2 + \sum_{i=1}^{n} \alpha(i) \cdot \left( S_{exp,i} - S_{sim,i} \right)^2. \tag{3}$$

The weights $\alpha(i)$ are normalized ($\sum_{i=1}^{n} \alpha(i) = n$) and equidistant ($\alpha(i+1) - \alpha(i) = \alpha(i) - \alpha(i-1)$ for all $i = 2, \ldots, n-1$), and chosen such that $10 \cdot \alpha(1) = \alpha(n)$. The used model was unchanged (Model B+), and the third strain generation was used for this experiment. The whole NMPC cycle procedure was executed as described for the previous experiment, starting with the glucose drop time prediction by the end of the batch phase and following with the first model parameter estimation and the first feeding rate calculation (for the estimated parameter and feeding rate values, see Table A13). In Figure 8a, the state of the cultivation at this first NMPC cycle is shown. In the following, eight NMPC cycles were performed. The results of each parameter estimation and the resulting calculated feed trajectory are listed in Table A14. Figure 8b shows the finished process. All the estimated parameters and the calculated feed trajectories for this experiment can be found in Section Appendix A.3 in Tables A12–A14.

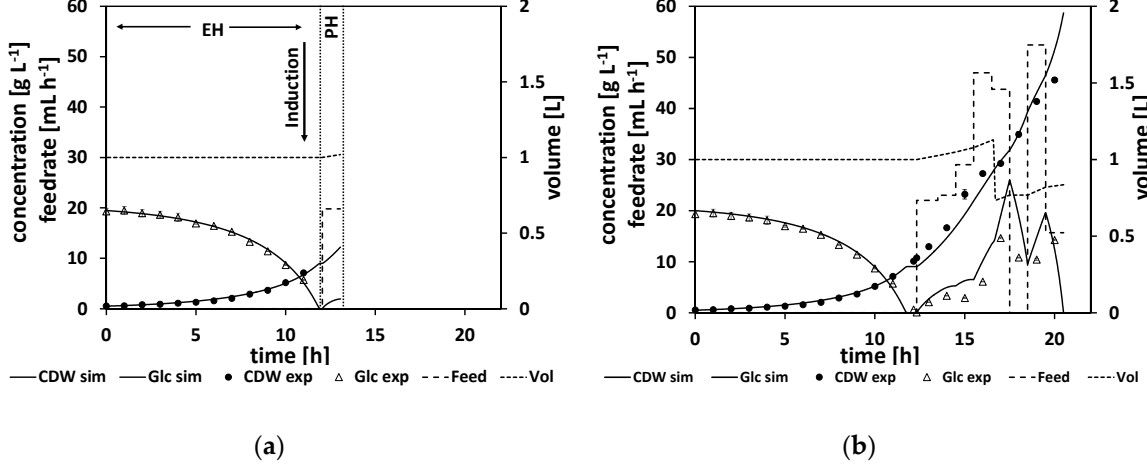

(**a**) 　　　　　　　　　　　　　　　　　　　　　(**b**)

**Figure 8.** Time course of the glucose and biomass concentrations for fed-batch 3. Left side of the vertical dotted lines indicates the estimation horizon (EH): symbols mark samples, the samples were measured as three technical replicates, the lines mark the adapted simulated cultivation course (Model B+), the dashed line shows the calculated feed profile, the dotted line marks the simulated reactor volume. Between the vertical dotted lines, indicating the prediction horizon (PH), the dashed line shows the optimized feed profile, the lines mark the extrapolated cultivation course, the dotted line marks the simulated reactor volume. (**a**) Prediction of the starting time point for the feed and calculation of the optimized first feed rate with the extrapolation starting time point $\tau = 12$ h. (**b**) Finished process. For the parameter estimation results and the feed optimization results, see Tables A12–A14. For model equations, see Table A11 or Table 2.

As before, a high biomass concentration of 45 g $L^{-1}$ was reached, which corresponds to an $OD_{600}$ of about 115. This concentration is similar to the concentrations reached in the previous cultivation, despite a new strain being used. Unfortunately, the measured glucose levels at the later stages of

the experiment were still too high, indicating an overfeeding of the NMPC algorithm and thereby an insufficient process control. Nonetheless, the simulation seems to fit the experimental data better compared to the last experiment (Figure 8b).

### 3.2.4. Experimental Setup 4-Strain 4–NMPC Fed-Batch Model C

As a result of the previous experiments and within the framework of the process development strategy (Section 1.2 step 3 → step 2), this time the model was adjusted accordingly (see the middle part of Figure 2) to take the findings of the previous experiments into account.

In detail, in order to slow down the simulated growth at later stages of the feed phase even more, a not-further-characterized growth-inhibiting metabolite was added to the model. The corresponding model is denoted Model C (see Table 2 or Table A15). The balance equations for the substrate (4) and the growth rate (9) were adjusted accordingly. The concentration of this metabolite starts to increase at the feed start and slows down the reaction rates as it accumulates. The inhibiting effect is larger as the metabolite concentration increases. As the metabolite is not characterized any further and especially not measured in any way, the initial metabolite concentration has to be predefined (see Table A16). The corresponding model parameters describing the metabolite and its growth-diminishing effect were received from a parameter adaptation of Model C to the previous experiment 3. The parameters values are listed in Table A16.

The actual experiment 4 was performed in the same way as the previous two experiments following the NMPC cycle procedure discussed in Section 2.2 and following the routine displayed in Figure 1. Figure 9a shows the first NMPC cycle, where the initial glucose drop time is calculated and a first parameter adaptation with a subsequent feeding rate is calculated (for the resulting parameter and feeding rate values, see Table A17), whereas in Figure 9b–d different NMPC cycles at later stages of the process are shown. Noteworthy in comparison to the previous experiments is the metabolite, which is presented by a dashed-dotted line. Its concentration starts to increase at the start of the feed phase. All the estimated parameters and the calculated feed trajectories for this experiment can be found in Appendix A.4 in Tables A16–A18.

The predicted drop time of about $\tau = 21.3$ h was in very good agreement with the time point of the DO signal increase at about $\tau = 21.25$ h (see Tables A17 and A18). A biomass concentration of 40 g $L^{-1}$ was reached, which corresponds to an $OD_{600}$ of about 103. This concentration is similar to the concentrations reached in previous cultivations. In general, compared to the previous strain generations, strain 4 grows slower.

In the beginning of the feed phase, the growth rate diminishes. As the first feeding rate is calculated by the extrapolation based on a model adapted to the data from the batch phase, this leads to an initial overfeed (Figure 9a). Figure 9b,c show how the NMPC algorithm reacts to this initial overfeed leading to a wavering feed profile.

In contrast to the former experiments, the NMPC kept the glucose at relatively low levels over the course of the fed-batch. As discussed earlier, this is desirable behavior for the NMPC algorithm, as ideally the NMPC algorithm should keep the glucose concentration relatively low at about $\sqrt{K_S I_S}$ (which is at about 1.6 g $L^{-1}$), which is the glucose level that would maximize the growth rate according to the model structure. Therefore, this experiment was successful. It can already be used to derive a standardized operation procedure by smoothening the feed profile.

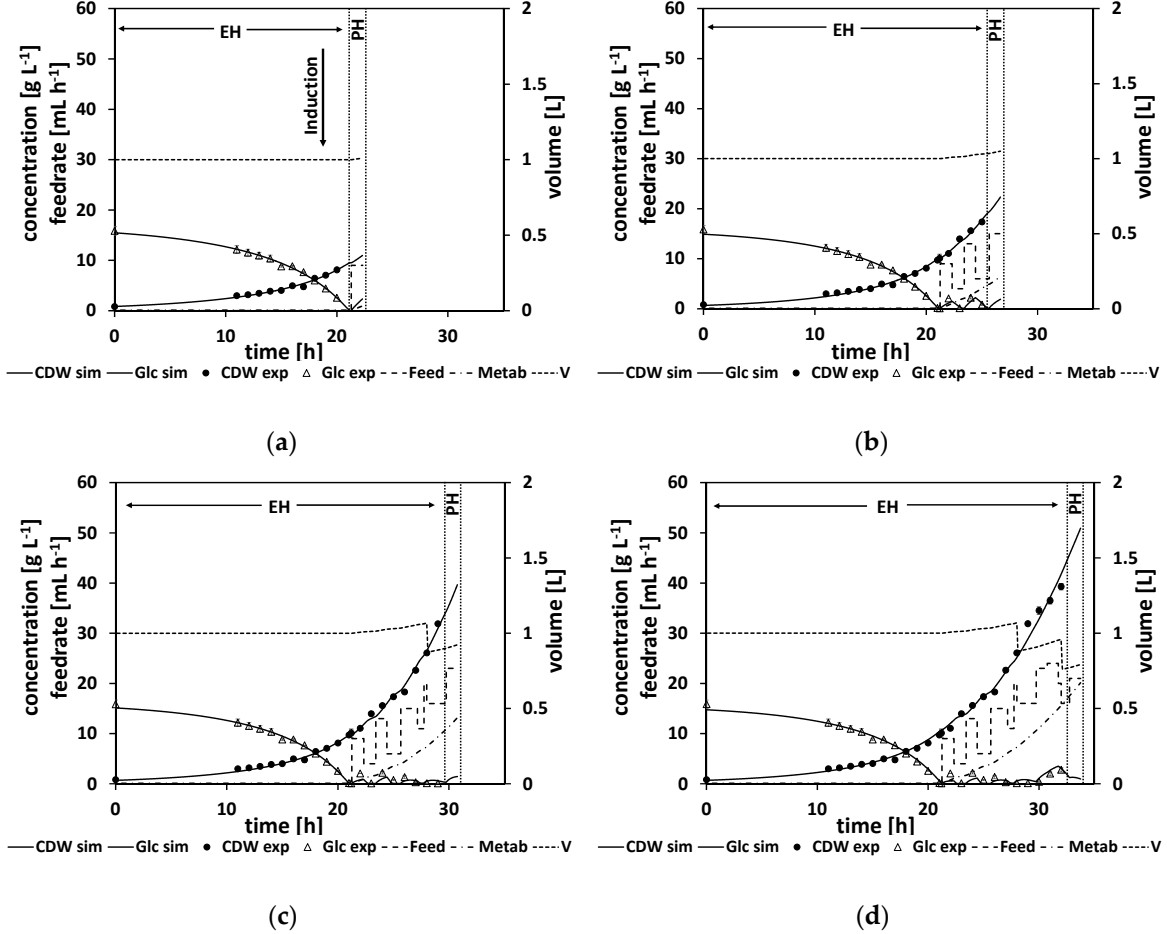

**Figure 9.** Time course of the glucose and biomass concentration for fed-batch 4. Left side of the vertical dotted lines indicates the estimation horizon (EH): symbols mark samples, the samples were measured as three technical replicates, the lines mark adapted the simulated cultivation course (Model B+), the dashed line shows the calculated feed profile, the dotted line marks the simulated reactor volume. Between the vertical dotted lines indicating the prediction horizon (PH): the dashed line shows the optimized feed profile, the lines mark the extrapolated cultivation course, the dotted line marks the simulated reactor volume. (**a**) Prediction of the starting time point for the feed and the calculation of the optimized first feed rate with extrapolation starting time point $\tau = 21$ h. (**b–d**) Calculation of the optimized feed rate for $\tau = 25.66$ h, $\tau = 29.75$ h, and $\tau = 32.75$ h. For the parameter estimation results and the feed optimization results, see Tables A16–A18. For the model equations, see Table A15 or Table 2.

### 3.2.5. Experiment 4-Strain 4-Fed-Batch Model D–Product Formation

For the last experiment 4, the product concentration of 2'-Fucosyllactose (2'-FL) was analyzed. As these analytics were performed after the actual experiment, the results could not be used for the actual NMPC algorithm. It is conceivable that, in future, the approaches for the discussed NMPC algorithm could be modified in order to maximize the product building instead of cell growth. However, this might lead to the same results as our current approach, as the following analysis suggests a growth-coupled product formation.

In order to simulate the product formation, the model was extended accordingly (see Table 2 or Table A19). The resulting model is denoted Model D. In Model D, the states and balance equations for the inducer and the product formation are added. The product formation rate $q_P$ was chosen to be proportional to the growth rate $\mu$, such that a growth-coupled product formation is simulated. The parameter estimation results are listed in Appendix A.5 in Tables A20–A22.

In Figure 10, the time course for the measured concentration of 2′-FL is shown as marks together with the simulation according to Model D. As the simulation with Model D shows a good agreement with the measured data, a growth-coupled product formation can be assumed.

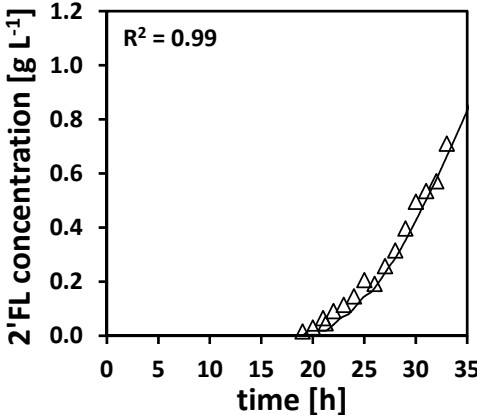

**Figure 10.** Time course of the product concentration 2′-FL for fed-batch 4: symbols mark the measured product concentration, the line marks adapted simulated cultivation course for the product (Model D). For the parameter estimation results, see Tables A20–A22. For model equations, see Table A19 or Table 2.

## 4. Conclusions

An NMPC for bioprocess fed-batch experiments with a step width of one hour has been established. This is a short time window for bioreactor processes regarding the necessary experimental effort for the at-line analytics for biomass and glucose. A strategic approach for the usage of the NMPC as a process development tool has been outlined as a workflow.

This workflow was discussed for a *C. glutamicum* fed-batch process, where a satisfying Standard Operating Procedure (SOP) can already be derived. Instead of developing the strain first and the process afterwards, as is convenient, it was possible to develop the process itself concurrently to the strain, with mostly one experiment per strain generation. Additionally, an acceptable mathematical model is at hand that might be used for further process optimization and control. This novel approach therefore provides a tool for fast process development.

The whole bioreactor control including the NMPC is realized in the control software to enable the transfer of the concept to other fermentation processes. Further studies may investigate processes with more complex kinetics. Furthermore, the usage of the control as a Digital Twin could be further assessed [57–59].

**Author Contributions:** Conceptualization, R.P., J.K., and K.M.S.; methodology, P.L., F.F., U.K., and H.Z.; software, P.L., K.M.S., and V.C.H.; validation, P.L., F.F., and J.M.; formal analysis, P.L., F.F., and J.M.; investigation, P.L., F.F., and J.M.; resources, R.P. and J.K.; data curation, J.M., V.C.H., and R.P.; writing—original draft preparation, P.L. and F.F.; writing—review and editing, R.P., J.M., J.K., and U.K.; visualization, P.L. and F.F.; supervision, R.P., J.K., and K.M.S.; project administration, R.P., J.K., and K.M.S.; funding acquisition, R.P., J.K., and K.M.S. All authors have read and agreed to the published version of the manuscript.

**Funding:** This study was partially funded by EFRE and the Free and Hanseatic City of Hamburg (grant number 51078424, IFB Hamburg).

**Acknowledgments:** We acknowledge support for the Open Access fees by Hamburg University of Technology (TUHH) in the funding program Open Access Publishing.

**Conflicts of Interest:** All authors P.L., F.F., U.K., H.Z., J.M., V.C.H., and R.P. do not have any conflict of interest. K.M.S. is the Managing Director of Ingenieurbüro Dr.-Ing. Schoop GmbH, J.K. is the managing director of GALAB Laboratories GmbH.

## Appendix A

**Table A1.** CgXII medium (modified from [49]) for the cultivation of *C. glutamicum* strains. Cultivations with pH control (pH 7) were performed with MOPS-free CgXII medium.

| Salt Solution | Concentration (g·L$^{-1}$) |
|---|---|
| $(NH_4)_2SO_4$ | 20 |
| Urea | 5 |
| MOPS | 42 |
| $MgSO_4 \cdot 7\,H_2O$ | 0.25 |
| $CaCl_2$ | 0.01 |
| $K_2HPO_4$ | 1 |
| $KH_2PO_4$ | 1 |

| C-Source | Concentration (g·L$^{-1}$) |
|---|---|
| D-Glucose | 20 |

| Vitamins | Concentration (g·L$^{-1}$) |
|---|---|
| Biotin | 0.0002 |
| Protocatechuic acid | 0.00003 |

| Trace Elements | Concentration (g·L$^{-1}$) |
|---|---|
| $FeSO_4 \cdot 7\,H_2O$ | 0.01 |
| $MnSO_4 \cdot H_2O$ | 0.01 |
| $ZnSO_4 \cdot 7\,H_2O$ | 0.001 |
| $CuSO_4$ | 0.0002 |
| $NiCl_2 \cdot 6\,H_2O$ | 0.00002 |

**Table A2.** List of symbols.

| | | |
|---|---|---|
| $\mu$ | specific growth rate | [h$^{-1}$] |
| $\mu_{max}$ | maximum specific growth rate biomass | [h$^{-1}$] |
| CDW | cell dry weight | [g L$^{-1}$] |
| $F_{in}$ | feed rate in | [L h$^{-1}$] |
| $F_{out}$ | feed rate out | [L h$^{-1}$] |
| GLC | glucose | [g L$^{-1}$] |
| $I_M$ | metabolite inhibition constant | [g L$^{-1}$] |
| $I_s$ | substrate inhibition constant | [g L$^{-1}$] |
| $K_L$ | monod constant inducer | [g L$^{-1}$] |
| $K_s$ | monod constant substrate | [g L$^{-1}$] |
| L | concentration inducer | [g L$^{-1}$] |
| M | metabolite concentration | [g L$^{-1}$] |
| $M_{init}$ | initial metabolite concentration | [g L$^{-1}$] |
| P | concentration product | [g L$^{-1}$] |
| q | production rate metabolite | [h$^{-1}$] |
| $q_{max}$ | maximum production rate metabolite | [h$^{-1}$] |
| $q_P$ | production rate product | [h$^{-1}$] |
| $q_{P,max}$ | maximum production rate product | [h$^{-1}$] |
| S | glucose concentration | [g L$^{-1}$] |
| $S_{init}$ | initial glucose concentration | [g L$^{-1}$] |
| $S_0$ | glucose concentration feed | [g L$^{-1}$] |
| $S_{esp,i}$ | experimental sample point substrate | [g L$^{-1}$] |
| $S_{sim,i}$ | simulated sample point substrate | [g L$^{-1}$] |
| t | time | [h] |
| V | volume | [L] |
| $V_{init}$ | initial volume | [L] |
| X | biomass concentration | [g L$^{-1}$] |
| $X_{init}$ | initial biomass concentration | [g L$^{-1}$] |
| $X_{exp,i}$ | experimental sample point biomass | [g L$^{-1}$] |
| $X_{sim,i}$ | simulated sample point biomass | [g L$^{-1}$] |
| $Y_{MS}$ | yield coefficient metabolite/substrate | [g g$^{-1}$] |
| $Y_{PS}$ | yield coefficient product/substrate | [g g$^{-1}$] |
| $Y_{PL}$ | yield coefficient product/inducer | [g g$^{-1}$] |
| $Y_{XS}$ | yield coefficient biomass/substrate | [g g$^{-1}$] |

*Appendix A.1 Strain 1–Batch and Fed-Batch Model A/B*

**Table A3.** Model equations for the initial batch experiment: Model A.

| *Balance equations* | |
| --- | --- |
| $\frac{dX}{dt} =$ | $\mu \cdot X - \frac{F}{V} \cdot X$ |
| $\frac{dS}{dt} =$ | $-\frac{1}{Y_{XS}} \cdot \mu \cdot X + \frac{F}{V} \cdot (S_0 - S)$ |
| $\frac{dV}{dt} =$ | $F_{in} - F_{out}$ |
| *Kinetic equations* | |
| $\mu =$ | $\mu_{max} \cdot \frac{S}{(S + K_s)}$ |

**Table A4.** Model parameters for the maximum specific growth rate biomass $\mu_{max}$, yield coefficient biomass/substrate $Y_{XS}$, and monod constant substrate $K_S$. Parameters were estimated by adaptation to batch experiments.

| $\mu_{max}[h^{-1}]$ | $Y_{XS}[g\ g^{-1}]$ | $K_S[g\ L^{-1}]$ |
| --- | --- | --- |
| 0.68 | 0.58 | 0.12 |

**Table A5.** Model equations of Model B.

| *Balance equations* | |
| --- | --- |
| $\frac{dX}{dt} =$ | $\mu \cdot X - \frac{F}{V} \cdot X$ |
| $\frac{dS}{dt} =$ | $-\frac{1}{Y_{XS}} \cdot \mu \cdot X + \frac{F}{V} \cdot (S_0 - S)$ |
| $\frac{dV}{dt} =$ | $F_{in} - F_{out}$ |
| *Kinetic equations* | |
| $\mu =$ | $\mu_{max} \cdot \frac{S}{(S + K_s)} \cdot \frac{I_S}{(S + I_S)}$ |

**Table A6.** Estimated initial concentrations for biomass $X_{init}$ and glucose $S_{init}$ and the estimated model parameters for the maximum specific growth rate biomass $\mu_{max}$, yield coefficient biomass/substrate $Y_{XS}$, monod constant substrate $K_S$, and substrate inhibition constant $I_S$. Adaptation to the fed-batch experiment.

| $X_{init}[g\ L^{-1}]$ | $S_{init}[g\ L^{-1}]$ | $\mu_{max}[h^{-1}]$ | $Y_{XS}[g\ g^{-1}]$ | $K_S[g\ L^{-1}]$ | $I_S[g\ L^{-1}]$ |
| --- | --- | --- | --- | --- | --- |
| 1.30 | 6.09 | 0.28 | 0.55 | 0.10 | 17.81 |

*Appendix A.2 Strain 2–NMPC Fed-Batch Model B+*

**Table A7.** Model equations for NMPC: Model B+.

| *Balance equations* | |
| --- | --- |
| $\frac{dX}{dt} =$ | $\mu \cdot X - \frac{F}{V} \cdot X$ |
| $\frac{dS}{dt} =$ | $-\frac{1}{Y_{XS}} \cdot \mu \cdot X + \frac{F}{V} \cdot (S_0 - S)$ |
| $\frac{dV}{dt} =$ | $F_{in} - F_{out}$ |
| *Kinetic equations* | |
| $\mu =$ | $\mu_{max} \cdot \frac{S}{(S + K_s)} \cdot \frac{I_S}{(S + I_S)}$ |

**Table A8.** Initial concentration volume $V_{init}$. Model parameters for the monod constant substrate $K_S$ and the substrate inhibition constant $I_S$. Parameters estimated by adaptation to previous experiments.

| $V_{init}[L]$ | $K_s[g\ L^{-1}]$ | $I_S[g\ L^{-1}]$ |
| --- | --- | --- |
| 1.2 | 0.1 | 25 |

**Table A9.** Estimated initial concentrations for biomass $X_{init}$ and glucose $S_{init}$ and the estimated model parameters for maximum specific growth rate biomass $\mu_{max}$ and yield coefficient biomass/substrate $Y_{XS}$ at process timepoint t. Calculated glucose drop time at $t_F$ and calculated optimized feed rate $F_{in}$ for timepoint $t_F$.

| t [h] | $X_{init}[g\ L^{-1}]$ | $S_{init}[g\ L^{-1}]$ | $\mu_{max,1}[h^{-1}]$ | $Y_{XS,1}[g\ g^{-1}]$ | $t_F[h]$ | $F_{in}[mL\ h^{-1}]$ |
|---|---|---|---|---|---|---|
| 8 | 0.21 | 19.94 | 0.68 | 0.54 | 8.49 | 42 |

**Remark A1.** *As the starting time point for the feed, the calculated drop time was not used, but rather an increase in the DO signal, indicating that the glucose dropped in the experiment. That is why the first tabulated value for $t_F$ in the following table can differ from the one above.*

**Table A10.** Estimated initial concentrations for biomass $X_{init}$ and glucose $S_{init}$ and the estimated model parameters for the maximum specific growth rate biomass $\mu_{max}$ and yield coefficient biomass/substrate $Y_{XS}$ (for batch phase index 1 and for fed-batch phase index 2) at process timepoint t. Calculated optimized feed rate $F_{in}$ for timepoint $t_F$.

| t [h] | $X_{init}[g\ L^{-1}]$ | $S_{init}[g\ L^{-1}]$ | $\mu_{max,1}[h^{-1}]$ | $Y_{XS,1}[g\ g^{-1}]$ | $\mu_{max,2}[h^{-1}]$ | $Y_{XS,2}[g\ g^{-1}]$ | $t_F[h]$ | $F_{in}[mL\ h^{-1}]$ |
|---|---|---|---|---|---|---|---|---|
| 8 | 0.21 | 19.94 | 0.68 | 0.54 | - | - | 9 | 42 |
| 9.5 | 0.47 | 20.28 | 0.54 | 0.59 | 0.27 | 0.21 | 10 | 42 |
| 10.5 | 0.30 | 19.87 | 0.62 | 0.61 | 0.35 | 0.31 | 11 | 44 |
| 11 | 0.51 | 20.26 | 0.53 | 0.60 | 0.32 | 0.30 | 12 | 40 |
| 12 | 0.30 | 19.82 | 0.62 | 0.62 | 0.31 | 0.28 | 13 | 55 |
| 13 | 0.31 | 20.09 | 0.62 | 0.60 | 0.31 | 0.28 | 14 | 0 |
| 14.5 | 0.30 | 19.91 | 0.61 | 0.60 | 0.31 | 0.29 | 15 | 84 |
| 15 | 0.29 | 20.49 | 0.62 | 0.60 | 0.31 | 0.29 | 15.5 | 23 |

**Remark A2.** *Due to experimental challenges, the cycle time for the parameter estimation and process optimization was shifting.*

*Appendix A.3 Strain 3–NMPC Fed-Batch Model B+ (Weighted Sample Points)*

**Table A11.** Model equations for NMPC: Model B+ (weighted sample points).

| *Balance equations* | |
|---|---|
| $\frac{dX}{dt} =$ | $\mu \cdot X - \frac{F}{V} \cdot X$ |
| $\frac{dS}{dt} =$ | $-\frac{1}{Y_{XS}} \cdot \mu \cdot X + \frac{F}{V} \cdot (S_0 - S)$ |
| $\frac{dV}{dt} =$ | $F_{in} - F_{out}$ |
| *Kinetic equations* | |
| $\mu =$ | $\mu_{max} \cdot \frac{S}{(S+K_s)} \cdot \frac{I_s}{(S+I_s)}$ |

**Table A12.** Initial concentration volume $V_{init}$. Model parameters for the monod constant substrate $K_S$ and substrate inhibition constant $I_S$. Parameters estimated by adaptation to previous experiment.

| $V_{init}[L]$ | $K_s[g\ L^{-1}]$ | $I_S[g\ L^{-1}]$ |
|---|---|---|
| 1 | 0.1 | 25 |

**Table A13.** Estimated initial concentrations for biomass $X_{init}$ and glucose $S_{init}$ and estimated model parameters for maximum specific growth rate biomass $\mu_{max}$ and yield coefficient biomass/substrate $Y_{XS}$ at process timepoint t. Calculated glucose drop time at $t_F$ and calculated optimized feed rate $F_{in}$ for timepoint $t_F$.

| t [h] | $X_{init}$[g L$^{-1}$] | $S_{init}$[g L$^{-1}$] | $\mu_{max,1}$[h$^{-1}$] | $Y_{XS,1}$[g g$^{-1}$] | $t_F$[h] | $F_{in}$[mL h$^{-1}$] |
|---|---|---|---|---|---|---|
| 11 | 0.41 | 19.70 | 0.38 | 0.44 | 12.05 | 20 |

**Remark A3.** *As the starting time point for the feed, the calculated drop time was not used, but rather an increase in the DO signal, indicating that glucose dropped in the experiment. That is why the first tabulated value for $t_F$ in the following table can differ from the one above.*

**Table A14.** Estimated initial concentrations for biomass $X_{init}$ and glucose $S_{init}$ and estimated model parameters for the maximum specific growth rate biomass $\mu_{max}$ and yield coefficient biomass/substrate $Y_{XS}$ (for batch phase index 1 and for fed-batch phase index 2) at process timepoint t. Calculated optimized feed rate $F_{in}$ for timepoint $t_F$.

| t [h] | $X_{init}$[g L$^{-1}$] | $S_{init}$[g L$^{-1}$] | $\mu_{max,1}$[h$^{-1}$] | $Y_{XS,1}$[g g$^{-1}$] | $\mu_{max,2}$[h$^{-1}$] | $Y_{XS,2}$[g g$^{-1}$] | $t_F$[h] | $F_{in}$[mL h$^{-1}$] |
|---|---|---|---|---|---|---|---|---|
| 11 | 0.41 | 19.70 | 0.38 | 0.44 | - | - | 12.33 | 20 (22) |
| 13 | 0.38 | 19.49 | 0.40 | 0.50 | 0.48 | 0.63 | 13.5 | 23 |
| 14 | 0.39 | 19.47 | 0.40 | 0.51 | 0.37 | 0.46 | 14.5 | 29 |
| 15 | 0.36 | 19.40 | 0.41 | 0.50 | 0.39 | 0.50 | 15.5 | 47 |
| 16 | 0.42 | 19.75 | 0.38 | 0.45 | 0.37 | 0.45 | 16.5 | 43 |
| 17 | 0.41 | 19.80 | 0.38 | 0.45 | 0.37 | 0.46 | 17.5 | 4 (0) |
| 18 | 0.41 | 20.32 | 0.39 | 0.43 | 0.37 | 0.45 | 18.5 | 52 |
| 19 | 0.41 | 20.23 | 0.39 | 0.43 | 0.37 | 0.46 | 19.5 | 16 |

**Remark A4.** *Due to a defective pump, it was not possible to set feed rates lower than 22 mL h$^{-1}$ (other than 0 mL h$^{-1}$). In brackets are the actually used feed rates.*

*Appendix A.4 Strain 4–NMPC Fed-Batch Model C*

**Table A15.** Model equations for NMPC. Model C.

*Balance equations*

$$\frac{dX}{dt} = \mu \cdot X - \frac{F}{V} \cdot X$$

$$\frac{dS}{dt} = -\frac{1}{Y_{XS}} \cdot \mu \cdot X + \frac{F}{V} \cdot (S_0 - S) - \frac{1}{Y_{MS}} \cdot q \cdot X$$

$$\frac{dM}{dt} = q \cdot X - \frac{F}{V} \cdot M$$

$$\frac{dV}{dt} = F_{in} - F_{out}$$

*Kinetic equations*

$$\mu = \mu_{max} \cdot \frac{S}{(S+K_s)} \cdot \frac{I_S}{(S+I_S)} \cdot \frac{I_M}{(M+I_M)}$$

$$q = q_{max} \cdot \frac{\mu}{\mu_{max}} \text{ with } q_{max} = 0 \text{ for } t < t_{Feed,Start}$$

**Table A16.** Initial concentrations for metabolite $M_{init}$ and volume $V_{init}$. Model parameters for monod constant substrate $K_S$, substrate inhibition constant $I_S$, yield coefficient metabolite/substrate $Y_{MS}$, maximum production rate metabolite $q_{max}$, and metabolite inhibition constant $I_M$. Parameters estimated by adaptation to previous experiment.

| $M_{init}[g\ L^{-1}]$ | $V_{init}[L]$ | $K_s[g\ L^{-1}]$ | $I_S[g\ L^{-1}]$ | $Y_{MS}[g\ g^{-1}]$ | $q_{max}$ $[h^{-1}]$ | $I_M$ $[g\ L^{-1}]$ |
|---|---|---|---|---|---|---|
| 0.1 | 1 | 0.1 | 25 | 977 | 0.1 | 51.8 |

**Table A17.** Estimated initial concentrations for biomass $X_{init}$ and glucose $S_{init}$ and estimated model parameters for the maximum specific growth rate biomass $\mu_{max}$ and yield coefficient biomass/substrate $Y_{XS}$ at process timepoint t. Calculated glucose drop time at $t_F$ and calculated optimized feed rate $F_{in}$ for timepoint $t_F$.

| t [h] | $X_{init}[g\ L^{-1}]$ | $S_{init}[g\ L^{-1}]$ | $\mu_{max,1}[h^{-1}]$ | $Y_{XS,1}[g\ g^{-1}]$ | $t_F[h]$ | $F_{in}[mL\ h^{-1}]$ |
|---|---|---|---|---|---|---|
| 20 | 0.77 | 15.63 | 0.17 | 0.56 | 21.3 | 9 |

**Remark A5.** *As the starting time point for the feed, the calculated drop time was not used, but rather an increase in the DO signal, indicating that the glucose dropped in the experiment. That is why the first tabulated value for $t_F$ in the following table can differ from the one above. In this experiment, the drop time was predicted quite accurately.*

**Table A18.** Estimated initial concentrations for biomass $X_{init}$ and glucose $S_{init}$ and the estimated model parameters for the maximum specific growth rate biomass $\mu_{max}$ and yield coefficient biomass/substrate $Y_{XS}$ (for batch phase index 1 and for fed-batch phase index 2) at process timepoint t. Calculated optimized feed rate $F_{in}$ for timepoint $t_F$.

| t [h] | $X_{init}[g\ L^{-1}]$ | $S_{init}[g\ L^{-1}]$ | $\mu_{max,1}[h^{-1}]$ | $Y_{XS,1}[g\ g^{-1}]$ | $\mu_{max,2}[h^{-1}]$ | $Y_{XS,2}[g\ g^{-1}]$ | $t_F[h]$ | $F_{in}[mL\ h^{-1}]$ |
|---|---|---|---|---|---|---|---|---|
| 20 | 0.77 | 15.63 | 0.17 | 0.56 | - | - | 21.25 | 9 |
| 22 | 0.68 | 15.33 | 0.18 | 0.60 | 0.18 | 0.60 | 22.33 | 4 |
| 23 | 0.78 | 15.79 | 0.17 | 0.57 | 0.24 | 0.69 | 23.42 | 13 |
| 24 | 0.60 | 15.00 | 0.19 | 0.64 | 0.19 | 0.61 | 24.42 | 6 |
| 25 | 0.60 | 15.02 | 0.19 | 0.63 | 0.20 | 0.58 | 25.66 | 15 |
| 26 | 0.82 | 15.51 | 0.16 | 0.60 | 0.19 | 0.52 | 27.17 | 11 |
| 27 | 0.74 | 15.47 | 0.17 | 0.60 | 0.21 | 0.52 | 27.75 | 20 |
| 28 | 0.68 | 15.28 | 0.18 | 0.61 | 0.22 | 0.53 | 28.75 | 16 |
| 29 | 0.62 | 15.24 | 0.18 | 0.60 | 0.24 | 0.55 | 29.75 | 23 |
| 30 | 0.65 | 15.44 | 0.18 | 0.59 | 0.24 | 0.54 | 30.75 | 24 |
| 31 | 0.65 | 15.12 | 0.18 | 0.64 | 0.22 | 0.51 | 31.75 | 20 |
| 32 | 0.61 | 14.86 | 0.19 | 0.69 | 0.21 | 0.48 | 32.75 | 21 |

*Appendix A.5 Strain 4-Fed-Batch Model D*

**Table A19.** Model equations for NMPC: Model D.

| *Balance equations* |
| --- |

$$\frac{dX}{dt} = \mu \cdot X - \frac{F}{V} \cdot X$$

$$\frac{dS}{dt} = -\frac{1}{Y_{XS}} \cdot \mu \cdot X + \frac{F}{V} \cdot (S_0 - S) - \frac{1}{Y_{MS}} \cdot q \cdot X - \frac{1}{Y_{PS}} \cdot q_P \cdot X$$

$$\frac{dM}{dt} = q \cdot X - \frac{F}{V} \cdot M$$

$$\frac{dL}{dt} = -\frac{1}{Y_{PL}} \cdot q_P \cdot X - \frac{F}{V} \cdot L$$

$$\frac{dP}{dt} = q_P \cdot X - \frac{F}{V} \cdot P$$

$$\frac{dV}{dt} = F_{in} - F_{out}$$

| *Kinetic equations* |
| --- |

$$\mu = \mu_{max} \cdot \frac{S}{(S+K_s)} \cdot \frac{I_S}{(S+I_S)} \cdot \frac{I_M}{(M+I_M)}$$

$$q = q_{max} \cdot \frac{\mu}{\mu_{max}} \text{ with } q_{max} = 0 \text{ for } t < t_{Feed,Start}$$

$$q_P = q_{P,max} \cdot \frac{\mu}{\mu_{max}} \cdot \frac{L}{(L+K_L)}$$

**Table A20.** Initial concentrations for metabolite $M_{init}$ and volume $V_{init}$. Model parameters for monod constant substrate $K_S$, substrate inhibition constant $I_S$, yield coefficient metabolite/substrate $Y_{MS}$, maximum production rate metabolite $q_{max}$, and metabolite inhibition constant $I_M$. Parameters estimated by adaptation to previous experiment.

| $M_{init}[g\ L^{-1}]$ | $V_{init}[L]$ | $K_s[g\ L^{-1}]$ | $I_S[g\ L^{-1}]$ | $Y_{MS}[g\ g^{-1}]$ | $q_{max}[h^{-1}]$ | $I_M[g\ L^{-1}]$ |
| --- | --- | --- | --- | --- | --- | --- |
| 0.1 | 1 | 0.1 | 25 | 977 | 0.1 | 51.8 |

**Table A21.** Estimated initial concentrations for biomass $X_{init}$ and glucose $S_{init}$ and the estimated model parameters for the maximum specific growth rate biomass $\mu_{max}$ and yield coefficient biomass/substrate $Y_{XS}$ (for batch phase index 1 and for fed-batch phase index 2).

| $X_{init}[g\ L^{-1}]$ | $S_{init}[g\ L^{-1}]$ | $\mu_{max,1}[h^{-1}]$ | $Y_{XS,1}[g\ g^{-1}]$ | $\mu_{max,2}[h^{-1}]$ | $Y_{XS,2}[g\ g^{-1}]$ |
| --- | --- | --- | --- | --- | --- |
| 0.61 | 14.86 | 0.21 | 0.69 | 0.29 | 0.48 |

**Table A22.** Estimated model parameters for the yield coefficient product/substrate $Y_{PS}$, yield coefficient product/inducer $Y_{PL}$, maximum production rate product $q_{P,max}$, and monod constant inducer $K_L$.

| $Y_{PS}[g\ g^{-1}]$ | $Y_{PL}[g\ g^{-1}]$ | $q_{P,max}[h^{-1}]$ | $K_L[g\ L^{-1}]$ |
| --- | --- | --- | --- |
| 0.026 | 1.565 | 0.004 | 0.100 |

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
