# Peer review of "NMPC-Based Workflow for Simultaneous Process and Model Development Applied to a Fed-Batch Process for Recombinant C. glutamicum"

_processes, doi:10.3390/pr8101313_

Round 1

Reviewer 1 Report

see attached 

Author Response

Dear Reviewer,

Thanks a lot for your valuable comments.

The manuscript was extensively revised and supplemented by an appendix showing numerical results in more detail. Especially the presentation of the NMPC, the associated process development workflow and its application to the C. glutamicum process were revised and thereby clarified.

In more detail:

Comment: However, the authors were using the model identification and model predictive control as two distinct moduls of the overall process and state estimation, and it seems that the features of moving horizon parameter and state estimation could be utilized as well (check Jim Rawlings papers on MPC and MHE). The model parameter estimation and process extrapolation is something that can be improved in the manuscript.

Answer: Papers form Jim Rawlings have been cited.

The model parameter estimation and process extrapolation has been re-written and hopefully improved. Regarding moving horizon parameter and state estimation - we indeed use this features. During fed-batch phase model parameter estimation and state estimation is repeated every hour (samples for determination of new data for cell density und substrate concentration are taken and analyzed every hour). A moving prediction horizon is used as well, as the future time course of the process is calculated for the next hour following parameter and state estimation at a certain time point.

In more detail:

  • In the introduction the presentation of the general workflow of the NMPC was re-worked and the presentation of the associated workflow for process development was clarified. By following this workflow, the process can be developed in parallel to the strain (instead of developing the process after strain development is finished, as common approaches suggest). Therefore, the whole development time is reduced. The presentation of the process development workflow was adjusted accordingly.
  • It was clarified that the model parameters for the NMPC are reiterated over the course of a fed-batch. The parameter adaptation and the following process optimization is repeated with respect to a fixed time interval (1 hour). Therefore, model parameter values are re-estimated every hour during fed-batch phase prior to the following feed profile calculation. To make this more clear an “estimation horizon” (EH) and a “prediction horizon” (PH) were introduced (see section 2.2 and section 3).

The procedure of a NMPC cycle for the C. glutamicum processs is explained in more detail (section 2.2). Especially the parameter estimation with regards to equation (1), order of magnitude for X and S, time intervals, used experimental data for parameter estimation, n (number of available sample points), local minima, convergence criteria, etc. was overworked and supplemented with numerical values (also in the appendi

Reviewer 2 Report

Refreshingly, this is well written, it reveals application to actual experimental data, and it offers a  practicable method.  These joys are strong reason to publish the article.  However, I object to the promotionalism style of writing – the authors use many adjectives and superlatives, overly promote the product name and company, and do not provide essential details on the methodology that others would need to be able to duplicate their methodology.  Substantial revision to the writing is needed.

I see a concept in Figure 1, and much claimed in the Abstract, and Sections 1. and 1.1.  At the end of Section 1.1 I read “This strategy aims to develop the process as fast as possible with respect to pre-defined evaluation criteria, e.g. maximized biomass concentration or product yield. Following this strategic approach it is possible to develop the strain and the process itself in parallel. This already shortens the required process development time tremendously.”  Those are all large claims with many adjectives and superlatives, but I do not see exactly how it can all be achieved.  This needs to be rewritten with more details about the methodology and less promotionalism.

The opening of Section 1.2 claims “The NMPC algorithm was completely implemented in the control software WinErs (provided by IB Schoop GmbH), which was used to control the process. As a result the whole NMPC algorithm can be automated. All in all, an NMPC-assisted strategy has been established, which allows the fast development, the characterization and the control of any given bioreactor process.”  Again, much claim with no supporting details.  This is reading like a marketing brochure, claiming “My product can do everything,” not an article for a scientific journal.  I think that this section also needs to be rewritten tempering commercialism and the claims of greatness.

I have done much related to model adaptation on-line (adjusting model coefficient values to best fit model to data) and then using the adapting model for control.  In Section 2.1 Step 1, they state, “In general, it is recommended to start with an unstructured, unsegregated model for its robustness and to keep computing times low.”  In my experience, starting with a structured model (a first-principles model) is best.  Unstructured models can become fully corrupted by noise.  First-principles models have a mathematical structure, a functional relationship that guides the variable relations, which simplifies the model adapting, and prevents absurdities of unstructured generic models.  The authors need to better explain their guidance.

Step 2 says, “the next step is to run a NMPC controlled fed batch in a bioreactor”.  In my experience nonlinear model predictive control chooses an optimal future sequence of control actions, one that optimize something.  At each sampling the model is corrected due to process-model-mismatch and the future MV sequence is recalculated.  At each sampling only the first MV in the future sequence is implemented.  Is this what the authors are doing, or something else?  What is the optimization objective value (for example to minimize the time to a particular reaction state, or to maximize annual production of product per batch? Much explanation is missing.  The authors need to explain what they mean by NMPC.

Also in Step 2 they claim “Success is reached, if the outcome of the fed-batch is actually in coincidence with the beforehand defined target, e.g. maximized biomass concentration or product yield, indicating that the process is already close to be the optimized process.”  But if the a priori hypothesized optimum value is not the true possibly achievable optimum, if it is a sub-optimal outcome, but the batch outcome matched the sub-optimal hypothesis, then how can they claim “success is reached”?  This claim needs to be explained.

In Step 4 they write “This procedure is repeated, until the process is optimized sufficiently and a standardized optimized procedure can be derived. At this point the NMPC control can still be used passively as monitoring tool for the process.“  How is “optimized sufficiently” determined?  Why would NMPC control now be used “passively”?  The process will continue to be subject to variation.  Each new batch will be slightly different from any prior batch, and need active NMPC control?  Much explaining is needed.

In the rules that I am accustomed to, it is acceptable to mention a product or a company or a country name once.  It seems that the authors mention “WinErs” and “IB Schoop GmbH” many times, violating rules about commercialism or promotionalism.  

Equation (1) has equal weighting on “biomass” and “glucose”.  Why?  To defend the choice of equal weighting in developing a model, the authors need to reveal dimensional units, the magnitudes, the uncertainty, and the relative importance to that variable on the model quality.  If, for instance, the product is glucose, the reasons to include biomass in the model fitting objective would be secondary, and the weighting should favor the glucose.  If, for instance, the two are equally important, but if the units on biomass are grams and the values are on the order of 1,000, and if the units on glucose are mg/L with values on the order of 1, then the square of the 1,000:1 values makes the biomass 10^6 times more important in model development than glucose.  Much clarification is needed.

In Equation (1), n needs to be defined.  Is it the number of data samples from the batch?  When does n start, when does n end, and is n the same for all batch runs?

In fitting model to data: Is the fitting only after the end of the batch, or progressively during the batch data evolution?  I assume NMPC control is active during a batch run that generates data.  My experience is that on-line adjustment of models confounds control.  What exactly do the authors do?  It is not clear.

Further, it is common to use prior batch data to adjust models or a batch recipe, and to use that model/recipe on the next batch.  But data are subject to noise and independent disturbances.  Even if the old model were perfect, the vagaries of the last batch will cause the model to change, leading to sub-optimal control on the next batch.  The Statistical Process Control term for taking action based on noise is tampering.  How do the authors temper the batch-to-batch model change?  Do they use averaging of the past 10 batches, incremental model coefficient value updating, statistical triggers to adjust a model coefficient value?  Explain.

Nelder-Mead is a fully acceptable optimization algorithm, but it can get stuck in a local optima.  The authors need to defend that there is only one minima to the least squares fitting. 

What are the initial trial solution values for the N-M optimization?

What is the convergence criteria?  The authors need to reveal that is was appropriately small relative to the effect on model functionality.

There are many control approaches that could be termed nonlinear and model-predictive. The authors need to explain theirs.  Is it truly horizon-predictive, constraint avoiding, control?  What is the control horizon, what is the reference trajectory, what is the optimizer? What exactly is the controller choosing?

Now it seems, in Equation (2) that the authors are using weighting factors on the regression objective function relative to late-batch data importance.  If this is what they recommend, then why distract the reader by showing Equation (1)?  At least explain with Equation (1) that it was a first attempt, that was subsequently replaced.  On Equation (2) I still have the same questions as I did on Equation (1) about units, magnitudes, and n.

I have many objections to what is written in the conclusion section.  The authors write:

  1. “A fast NMPC for bioprocess fed-batch experiments with a step width of one hour has been established.” “Fast” was never defended.  “Fast” with respect to what?
  2. “This is a short time window for bioreactor processes regarding the necessary experimental effort.” What is the “this” that is short? Does experimental effort just mean the batch run-time, or does it include the material and equipment setup and post experiment clean-up?
  3. “This workflow was discussed on a C. glutamicum fed-batch process, where a satisfying Standard Operating Procedure (SOP) could already be derived.” Where was the SOP discussed in the manuscript?  What makes it “satisfying”?
  4. “It was possible to develop the process itself parallel to the strain with mostly one experiment per strain generation and thereby to reduce the process development time and the effort was reduced tremendously.” What does “parallel to the strain” mean?  Does it mean while the batch is being run, the model is being developed?  Does “process development” mean the design of the process or the least squares fitting of the model to data?  Yes, one can get a best fit model to one batch run, but is that model representative of all batches or just a match to the experimental vagaries of the single batch?  If the development time is “reduced” and “reduced tremendously” what is it compared to, and where is the data to support this adjective and its superlative?
  5. “Additionally, a proven mathematical model is ad hand, that can be used for furthermore process optimization and control.” I think it should be “at”, not “ad”.  Where is the evidence to support that the model is “proven” and that it can be “used for process optimization”?  That is speculation.  I believe it to be so, but the authors offer no evidence.
  6. “This novel approach therefore provides a tool for fast process development.” Where is the evidence for the adjective “fast”, and fast relative to what?
  7. “The whole bioreactor control inclusive the NMPC is realized in the control software WinErs.” Inappropriate commercialism.
  8. “This makes the whole concept workable for nearly any given process.” No evidence provided to support this claim. It is speculation.
  9. “Further studies will investigate on processes with more complex kinetics. Furthermore, the usage of the control as Digital Twin will be assessed.” It seems the authors think that they can predict the future! Instead of “will”, do they mean that their “plan” is to investigate and to assess?

Author Response

Dear Reviewer,

Thanks a lot for your valuable comments.

The manuscript was extensively revised and supplemented by an appendix showing numerical results in more detail. Especially the presentation of the NMPC, the associated process development workflow and its application to the C. glutamicum process were revised and thereby clarified. In more detail:

Comment: Refreshingly, this is well written, it reveals application to actual experimental data, and it offers a practicable method. These joys are strong reason to publish the article. However, I object to the promotionalism style of writing – the authors use many adjectives and superlatives, overly promote the product name and company, and do not provide essential details on the methodology that others would need to be able to duplicate their methodology. Substantial revision to the writing is needed.

I see a concept in Figure 1, and much claimed in the Abstract, and Sections 1. and 1.1. At the end of Section 1.1 I read “This strategy aims to develop the process as fast as possible with respect to pre-defined evaluation criteria, e.g. maximized biomass concentration or product yield. Following this strategic approach it is possible to develop the strain and the process itself in parallel. This already shortens the required process development time tremendously.” Those are all large claims with many adjectives and superlatives, but I do not see exactly how it can all be achieved. This needs to be rewritten with more details about the methodology and less promotionalism.

The opening of Section 1.2 claims “The NMPC algorithm was completely implemented in the control software WinErs (provided by IB Schoop GmbH), which was used to control the process. As a result the whole NMPC algorithm can be automated. All in all, an NMPC-assisted strategy has been established, which allows the fast development, the characterization and the control of any given bioreactor process.” Again, much claim with no supporting details. This is reading like a marketing brochure, claiming “My product can do everything,” not an article for a scientific journal. I think that this section also needs to be rewritten tempering commercialism and the claims of greatness.

Answer: The manuscript was re-worked regarding the promotional style of writing and commercialism (section 1 and section 2.1)

Comment: I have done much related to model adaptation on-line (adjusting model coefficient values to best fit model to data) and then using the adapting model for control. In Section 2.1 Step 1, they state, “In general, it is recommended to start with an unstructured, unsegregated model for its robustness and to keep computing times low.” In my experience, starting with a structured model (a first-principles model) is best. Unstructured models can become fully corrupted by noise. First-principles models have a mathematical structure, a functional relationship that guides the variable relations, which simplifies the model adapting, and prevents absurdities of unstructured generic models. The authors need to better explain their guidance.

Answer: We followed the classification of models common for microbial fermentation processes (C. González-Figueredo, R.A. Flores-Estrella, O.A. Rojas-Rejón (2018) Fermentation: Metabolism, Kinetic Models, and Bioprocessing. DOI:10.5772/intechopen.82195). Simple and unstructured (unsegregated) kinetic growth models represent, in a simple global point of view, the metabolic behavior of the biomass cell production. Mainly, mathematical descriptions for microbial growth kinetics in fermentation processes are based on semi-empirical observations. From simple experimental data, information to represent cellular growth with unstructured kinetic models can be obtained. This was explained in the text (see section 1.2).

Comment: Step 2 says, “the next step is to run a NMPC controlled fed batch in a bioreactor”. In my experience nonlinear model predictive control chooses an optimal future sequence of control actions, one that optimize something. At each sampling the model is corrected due to processmodel-mismatch and the future MV sequence is recalculated. At each sampling only the first MV in the future sequence is implemented. Is this what the authors are doing, or something else? What is the optimization objective value (for example to minimize the time to a particular reaction state, or to maximize annual production of product per batch? Much explanation is missing. The authors need to explain what they mean by NMPC.

Answer: In the introduction the presentation of the general workflow of the NMPC (section 1.2) was re-worked and the presentation of the associated workflow for process development was clarified. By following this workflow the process can be developed in parallel to the strain (instead of developing the process after the strain development is finished, as common approaches suggest). Therefore, the whole development time is reduced. The presentation of the process development workflow was adjusted accordingly.

The NMPC controller is used to optimize the feeding rate within the coming time window, such that a maximized biomass concentration at the end of an extrapolated one hour time window is reached (see section 2.2.2).

Comment: Also in Step 2 they claim “Success is reached, if the outcome of the fed-batch is actually in coincidence with the beforehand defined target, e.g. maximized biomass concentration or product yield, indicating that the process is already close to be the optimized process.”  But if the a priori hypothesized optimum value is not the true possibly achievable optimum, if it is a sub-optimal outcome, but the batch outcome matched the sub-optimal hypothesis, then how can they claim “success is reached”?  This claim needs to be explained.

In Step 4 they write “This procedure is repeated, until the process is optimized sufficiently and a standardized optimized procedure can be derived. At this point the NMPC control can still be used passively as monitoring tool for the process.“  How is “optimized sufficiently” determined?  Why would NMPC control now be used “passively”?  The process will continue to be subject to variation.  Each new batch will be slightly different from any prior batch, and need active NMPC control?  Much explaining is needed.

Answer: It was clarified that the model parameters for the NMPC are re-iterated over the course of the fed-batch phase. The parameter adaptation and the following process optimization is repeated with respect to a fixed time interval (of 1 hour). Therefore, model parameter values are re-estimated every hour during fed-batch phase prior to the following feed profile calculation. To make this more clear, an “estimation horizon” (EH) and a “prediction horizon” (PH) were introduced (see section 2.2 and section 3).

Comment: In the rules that I am accustomed to, it is acceptable to mention a product or a company or a country name once.  It seems that the authors mention “WinErs” and “IB Schoop GmbH” many times, violating rules about commercialism or promotionalism. 

Answer: The manuscript was reworked regarding the promotional style of writing and commercialism (section 1 and section 2.1)

Comment: Equation (1) has equal weighting on “biomass” and “glucose”.  Why?  To defend the choice of equal weighting in developing a model, the authors need to reveal dimensional units, the magnitudes, the uncertainty, and the relative importance to that variable on the model quality.  If, for instance, the product is glucose, the reasons to include biomass in the model fitting objective would be secondary, and the weighting should favor the glucose.  If, for instance, the two are equally important, but if the units on biomass are grams and the values are on the order of 1,000, and if the units on glucose are mg/L with values on the order of 1, then the square of the 1,000:1 values makes the biomass 10^6 times more important in model development than glucose.  Much clarification is needed.

Answer: The procedure of a NMPC cycle for the C. glutamicum processs is explained in more detail (section 2.2). Especially the parameter estimation with regards to equation (1), order of magnitude for X and S, time intervals, used experimental data for parameter estimation, n (number of available sample points), local minima, convergence criteria, etc. was overworked and supplemented with numerical values (also in the appendix). The same points were clarified for calculation of the feed profile.

Comment: In Equation (1), n needs to be defined.  Is it the number of data samples from the batch?  When does n start, when does n end, and is n the same for all batch runs?

Answer: n indicates the number of available sample points and increases during the fed-batch in each optimization circle by one.  

Comment: In fitting model to data: Is the fitting only after the end of the batch, or progressively during the batch data evolution?  I assume NMPC control is active during a batch run that generates data.  My experience is that on-line adjustment of models confounds control.  What exactly do the authors do?  It is not clear.

Answer: The presentation of the results in section 3 was completely re-worked and extended with the emphasis to clarify the followed workflow. Here, also the application of the NMPC is explained in more detail by discussing the NMPC with regards to its estimation horizon and prediction horizon in-depth. Numerical values are added in the appendix. The progression of the experiments over the strain generations and the corresponding model extensions are discussed in more detail. It was highlighted that this progression was performed by following the process development workflow.   

Comment: Further, it is common to use prior batch data to adjust models or a batch recipe, and to use that model/recipe on the next batch.  But data are subject to noise and independent disturbances.  Even if the old model were perfect, the vagaries of the last batch will cause the model to change, leading to sub-optimal control on the next batch.  The Statistical Process Control term for taking action based on noise is tampering.  How do the authors temper the batch-to-batch model change?  Do they use averaging of the past 10 batches, incremental model coefficient value updating, statistical triggers to adjust a model coefficient value?  Explain.

Answer: The reviewer is correct. Batch-to-batch model changes and noise are important aspects. Unfortunately, all the techniques for statistical evaluation of the model parameters require additional experiments and/or sophisticated mathematical approaches (e.g. Monte Carlo) are very time consuming. This contradicts the goal of fast and efficient process development. Therefore, a more detailed evaluation can only be performed after the described first process design.  

Comment: Nelder-Mead is a fully acceptable optimization algorithm, but it can get stuck in a local optima.  The authors need to defend that there is only one minima to the least squares fitting.

What are the initial trial solution values for the N-M optimization?

What is the convergence criteria?  The authors need to reveal that is was appropriately small relative to the effect on model functionality.

Answer: The procedure of a NMPC cycle for the C. glutamicum processs is explained in more detail (section 2.2). Especially the parameter estimation with regards to equation (1), order of magnitude for X and S, time intervals, used experimental data for parameter estimation, n (number of available sample points), local minima, convergence criteria, etc. was overworked and supplemented with numerical values (also in the appendix). The same points were clarified for the feed profile calculation.

There are many control approaches that could be termed nonlinear and model-predictive. The authors need to explain theirs.  Is it truly horizon-predictive, constraint avoiding, control?  What is the control horizon, what is the reference trajectory, what is the optimizer? What exactly is the controller choosing?

Answer: In the introduction and in section 2.2 the presentation of the general workflow of the NMPC was re-worked and the presentation of the associated workflow for process development was clarified.

Comment: Now it seems, in Equation (2) that the authors are using weighting factors on the regression objective function relative to late-batch data importance.  If this is what they recommend, then why distract the reader by showing Equation (1)?  At least explain with Equation (1) that it was a first attempt, that was subsequently replaced.  On Equation (2) I still have the same questions as I did on Equation (1) about units, magnitudes, and n.

Answer: The equations have been explained more in detail. Compare section 2.2.1 and 2.2.2. Also more detailed explanations of magnitudes for X and S as well as for n (number of available sample points) have been added.

Comment: I have many objections to what is written in the conclusion section.  The authors write:

  1. “A fast NMPC for bioprocess fed-batch experiments with a step width of one hour has been established.” “Fast” was never defended. “Fast” with respect to what?

Answer: “Fast” in this respect refers to the duration of the fermentation process (compare Fig. 6). As shown in Fig. 4, from the time interval of 1 h approx. 20 min are consumed by sampling and analysis and 10 min by parameter estimation and feed profile calculation. According to the available literature, NMPC (mostly OLFO-controller) have been mainly used for slower fermentation processes (e.g. mammalian cell cultures lasting up to approx. 1 week).

  1. “This is a short time window for bioreactor processes regarding the necessary experimental effort.” What is the “this” that is short? Does experimental effort just mean the batch run-time, or does it include the material and equipment setup and post experiment clean-up?

Answer: see above. It just refers to the batch / fed-batch run time.

  1. “This workflow was discussed on a C. glutamicum fed-batch process, where a satisfying Standard Operating Procedure (SOP) could already be derived.” Where was the SOP discussed in the manuscript? What makes it “satisfying”?

Answer: The feed profile worked out for strain 4 (Fig. 4) can be used as first approach for an SOP.

  1. “It was possible to develop the process itself parallel to the strain with mostly one experiment per strain generation and thereby to reduce the process development time and the effort was reduced tremendously.” What does “parallel to the strain” mean? Does it mean while the batch is being run, the model is being developed?  Does “process development” mean the design of the process or the least squares fitting of the model to data?  Yes, one can get a best fit model to one batch run, but is that model representative of all batches or just a match to the experimental vagaries of the single batch?  If the development time is “reduced” and “reduced tremendously” what is it compared to, and where is the data to support this adjective and its superlative?

Answer: The introduction was re-worked in this respect. The development of bioprocesses can be time-consuming and cost-intensive. Usually, first the production strain for expression of the target molecule is created using molecular biology and recombinant DNA technology. After selection of a promising production strain, the actual process development for production of large quantities of the target molecule starts. By this step-by-step approach usually a lot of time is wasted.

With the suggested NMPC-based strategy it was possible to develop the fed batch process with just one fed-batch experiment per strain. The 4 strains were at hand mostly in time intervals of 4 – 6 weeks.

  1. “Additionally, a proven mathematical model is ad hand, that can be used for furthermore process optimization and control.” I think it should be “at”, not “ad”. Where is the evidence to support that the model is “proven” and that it can be “used for process optimization”?  That is speculation.  I believe it to be so, but the authors offer no evidence.

Answer: “ad hand” was corrected.

With respect to the model the sentence was changed as follows: “Additionally, a acceptable mathematical model is at hand, that might be used for furthermore process optimization and control.”

  1. “This novel approach therefore provides a tool for fast process development.” Where is the evidence for the adjective “fast”, and fast relative to what?

Answer: The sentence was deleted

  1. “The whole bioreactor control inclusive the NMPC is realized in the control software WinErs.” Inappropriate commercialism.

Answer: Has been changed, see above.

  1. “This makes the whole concept workable for nearly any given process.” No evidence provided to support this claim. It is speculation.

Answer: The sentence was changed as follows: “The whole bioreactor control including the NMPC is realized in the control software to enable transfer of the concept to other fermentation processes.”

  1. “Further studies will investigate on processes with more complex kinetics. Furthermore, the usage of the control as Digital Twin will be assessed.” It seems the authors think that they can predict the future! Instead of “will”, do they mean that their “plan” is to investigate and to assess?

Answer: The sentence was changed as follows: “Further studies may investigate on processes with more complex kinetics. Furthermore, the usage of the control as Digital Twin could be further assessed.

Reviewer 3 Report

General comments:

This study is interesting but the presentation could be largely improved.

The NMPC is not necessary for model development. It is only a parameter estimation  which requires only a nonlinear optimizer whereas NMPC is useful only for control. Furthermore, it is not certain that a great difference would exist if MPC was used instead of NMPC in the control part given the prediction horizon which is not large.

There is no reason to write "the potency of the NMPC algorithm as a powerful tool for process development tool and model generation is
demonstrated" as the model generation differs from NMPC action.

This manuscript should be completely revised.

Main remarks:
In the abstract, I consider that there is a confusion when the authors write
"The NMPC algorithm is capable of improving various process parameters like product yield and biomass concentration". The product yield and biomass concentration are not parameters, they are states for the dynamic model.

I consider that the sentence
"the NMPC algorithm for process development developed here is based on prior studies of [6, 9, 41] and consists of an identification part, which estimates the states and parameters of the process model using the available experimental data, and an optimization part, which calculates an e.g. optimal control feed trajectory" reveals a problem  of the present study, i.e. the confusion between the identification and the control.

Equation (1): the drawback of this equation is that, if the order of magnitude  of X and S is not the same, one term has a larger weight than the other one in the objective function.

Table 2: the equations for the models could be presented without The table, but as usual equations in any article. It would be clearer as, in particular, the vertical alignment with B, C is not clear.

Figure 5: rather than this quality of fit which is presented, I prefer
to know the confidence intervals of the parameters.

There is a Figure at end of page 8 and another figure at beginning of page 9 which seem to be the same.

Section 3.2.2: the explanation about the vertical dashed lines is not clear.
If Figures 7a to 7d represent the evolution at different instants, this is not explained in the text, nor in the captions.
In my opinion, only Figure 7d is useful, but I would have better explained
what is done at every calculation time, i.e. the calculation of the parameters of the model which does not require NMPC, and the calculation of the future feed rate which is based on NMPC.
A large part of the article could be improved by simply separating
the notions of parameter estimation and control and by giving the different time increments for sampling, for parameter estimation, for control to gether with the control and prediction horizons.
The sentence "show the cultivation courses at later stages of the experiment" is not clear (what means "later"? This is often used in the following without more clarity.)

Equation (2): the explanation about the weights is awkward. The authors have tried to introduce a forgetting factor. Nevertheless, my previous remark about eq.(1) remains valid.

Figure 8: the remark about Figure 7 remains also valid. Only Figure 8d is useful.

It seems that the successive improvements of the model (from Figs. 7 to 10) are obtained by an a priori knowledge of what should be the best result which is not obvious to a reader not familiar with the bioreaction.
It seems to me that the main point is the product yield. This does not appear clearly in the discussion.

Minor errors:

In general, use commas when necessary to render the text more understandable.
For example, it should be :
- line 17: "... NMPC algorithm, the process ..."
- line 19: "This way, the number ..."
- line 27: "... algorithm, process development ..."
- line 46: "Usually, ..."
- line 49: "For this, ..."
- lines ...
- line 248: "... with the NMPC algorithm, the experimental ..."
- line 280: "All in all, the NMPC ..."
- line 329, "In Figure 10, the time ..."
- line 331: "... data, a growth coupled ..."

line 180: correct as "cultivation"

line 341: correct "ad hand"

line 343: replace "inclusive" by "including"

Author Response

Dear Reviewer,

Thanks a lot for your valuable comments.

The manuscript was extensively revised and supplemented by an appendix showing numerical results in more detail. Especially the presentation of the NMPC, the associated process development workflow and its application to the C. glutamicum process were revised and thereby clarified. The manuscript was re-worked regarding the promotional style of writing and commercialism (section 1 and section 2.1)

In the introduction the presentation of the general workflow of the NMPC was reworked and the presentation of the associated workflow for process development was clarified. By following this workflow the process can be developed in parallel to the strain (instead of developing the process after the strain development is finished, as common approaches suggest). Therefore, the whole development time is reduced. The presentation of the process development workflow was adjusted accordingly.

In the presentation of the workflow, it was explained, what an unstructured, unsegregated model means in the context of microbial processes.

  1. González-Figueredo, R.A. Flores-Estrella, O.A. Rojas-Rejón (2018) Fermentation: Metabolism, Kinetic Models, and Bioprocessing. DOI:10.5772/intechopen.82195

It was clarified that the model parameters for the NMPC are reiterated over the course of a fed-batch. The parameter adaptation and the following process optimization is repeated with respect to a fixed time interval (of 1 hour). Therefore, model parameter values are re-estimated every hour during fed-batch phase prior to the following feed profile calculation. To make this more clear an “estimation horizon” (EH) and a “prediction horizon” (PH) were introduced (see section 2.2 and section 3).

In more detail:

Comment: In the abstract, I consider that there is a confusion when the authors write

"The NMPC algorithm is capable of improving various process parameters like product yield and biomass concentration". The product yield and biomass concentration are not parameters, they are states for the dynamic model.

Answer: The sentence has been changed: “The NMPC algorithm is capable of improving various process states like product yield and biomass concentration.”

Comment: I consider that the sentence "the NMPC algorithm for process development developed here is based on prior studies of [6, 9, 41] and consists of an identification part, which estimates the states and parameters of the process model using the available experimental data, and an optimization part, which calculates an e.g. optimal control feed trajectory" reveals a problem  of the present study, i.e. the confusion between the identification and the control.

Answer: The description of the NMPC and the workflow of process development have been revised (section 1.1 and 1.2).

Comment: Equation (1): the drawback of this equation is that, if the order of magnitude of X and S is not the same, one term has a larger weight than the other one in the objective function.

Answer: The procedure of a NMPC cycle for the C. glutamicum processs is explained in more detail (section 2.2). Especially the parameter estimation with regards to equation (1), order of magnitude for X and S, time intervals, used experimental data for parameter estimation, n (number of available sample points), local minima, convergence criteria, etc. was overworked and supplemented with numerical values (also in the appendix). The same points were clarified for the feed profile calculation.

The following was written in section 2.2.1: “The objective of the control is to maximize the biomass concentration by optimizing the glucose feeding rate. Therefore, these two states are chosen to be the relevant states for the model parameter adaptation. The reason glucose concentration and biomass concentration were chosen equally weighted for model fitting are the following: First of all the biomass concentration and glucose concentration samples were taken in constant time intervals (one hour), such that a weighting regarding differing time gaps is not needed. Consequently, the number of available data points for each are the same (denoted by n). Secondly, both are of the same order of magnitude varying from 1 to 50 g L-1. Therefore, it is sufficient to use absolute deviations of instead of for example relative deviations. For other processes the formula can be adjusted accordingly.”

Comment: Table 2: the equations for the models could be presented without the table, but as usual equations in any article. It would be clearer as, in particular, the vertical alignment with B, C is not clear.

Answer: For clarification, Table 2 was explained more in detail in the main text and the models used for the 4 different strains are summarized in the supplementary material. We hope that helps to understand the stepwise modification of the model.

Comment: Figure 5: rather than this quality of fit which is presented, I prefer to know the confidence intervals of the parameters.

Answer: The coefficient of determination R2 is given. Unfortunately, all the techniques for statistical evaluation of the model parameters require additional experiments and/or sophisticated mathematical approaches (e.g. Monte Carlo) are very time consuming. This contradicts the goal of fast and efficient process development. Therefore, a more detailed evaluation can only be performed after the described first process design.  

Comment: There is a Figure at end of page 8 and another figure at beginning of page 9 which seem to be the same.

Answer: Thanks, this was a misprint.

Comment: Section 3.2.2: the explanation about the vertical dashed lines is not clear.

If Figures 7a to 7d represent the evolution at different instants, this is not explained in the text, nor in the captions.

In my opinion, only Figure 7d is useful, but I would have better explained what is done at every calculation time, i.e. the calculation of the parameters of the model which does not require NMPC, and the calculation of the future feed rate which is based on NMPC.

Answer: The presentation of the results in section 3 was completely reworked and extended with the emphasis to clarify the followed workflow and its advantages. Here, also the application of the NMPC is explained in more detail by discussing the NMPC with regards to its estimation horizon and prediction horizon in-depth. Numerical values are added in the appendix. The progression of the experiments over the strain generations and the corresponding model extensions are discussed in more detail. It was highlighted that this progression was performed by following the process development workflow.  

Comment: A large part of the article could be improved by simply separating the notions of parameter estimation and control and by giving the different time increments for sampling, for parameter estimation, for control together with the control and prediction horizons.

The sentence "show the cultivation courses at later stages of the experiment" is not clear (what means "later"? This is often used in the following without more clarity.)

Answer: “Later” refers to the last hours of the fed-batch process (e.g. the last 2-3 hours, were cell growth decreases).

Comment: Equation (2): the explanation about the weights is awkward. The authors have tried to introduce a forgetting factor. Nevertheless, my previous remark about eq.(1) remains valid.

Answer: The equations have been explained more in detail. Compare section 2.2.1 and 2.2.2.

Comment: Figure 8: the remark about Figure 7 remains also valid. Only Figure 8d is useful.

Answer: We have reduced Fig. 8 to two time points (“end of batch / start of fed-batch phase” and “end of cultivation”)

Comment: It seems that the successive improvements of the model (from Figs. 7 to 10) are obtained by an a priori knowledge of what should be the best result which is not obvious to a reader not familiar with the bioreaction.

Answer: The step-wise-improvement of the model has been described more in detail, especially appropriate models from literature [7, 48-54] were cited. The terms used in the different models are common for unsegregated, unstructured models for bioreactions.

Comment: It seems to me that the main point is the product yield. This does not appear clearly in the discussion.

Answer: This is correct. But we focused on biomass concentration in the first place due to the following reasons. First, just the last strain (strain 4) was able to express sufficient amounts of the product (compare table 1). Second, analytics of the product is very time consuming and not possible “at line”, parallel to the running fed-batch process as for the biomass concentration.

Minor errors: In general, use commas when necessary to render the text more understandable.

For example, it should be :

- line 17: "... NMPC algorithm, the process ..." - done

- line 19: "This way, the number ..." – the sentence has been changed

- line 27: "... algorithm, process development ..." - done

- line 46: "Usually, ..." - done

- line 49: "For this, ..." – the sentence has been changed

- line 248: "... with the NMPC algorithm, the experimental ..." - done

- line 280: "All in all, the NMPC ..." - done

- line 329, "In Figure 10, the time ..." - done

- line 331: "... data, a growth coupled ..."

- line 180: correct as "cultivation" - done

- line 341: correct "ad hand" - done

- line 343: replace "inclusive" by "including" – done

Round 2

Reviewer 2 Report

I appreciate the author’s edits and additions.  I believe that I now fully understand what they are offering as better practice. 

Some very minor issues: 

On Line 15, the first sentence of the abstract, “were” should be “where”.

On Lines 119, 304, 328 there are reference errors.

I still object to their use of superlatives as unsupported claims.  “drastically” needs to be qualified in the abstract.

The authors should carefully inspect the manuscript.

There is no need for another review.